# Deep generative AI models analyzing circulating orphan non-coding RNAs enable detection of early-stage lung cancer

Mehran Karimzadeh [1,7], Amir Momen-Roknabadi[1,7], Taylor B. Cavazos[1,7], Yuqi Fang[1], Nae-Chyun Chen[1], Michael Multhaup[1], Jennifer Yen[1], Jeremy Ku[1], Jieyang Wang[1], Xuan Zhao[1], Philip Murzynowski[1], Kathleen Wang[1], Rose Hanna[1], Alice Huang[1], Diana Corti[1], Dang Nguyen[1], Ti Lam[1], Seda Kilinc[1], Patrick Arensdorf[1], Kimberly H. Chau[1], Anna Hartwig[1], Lisa Fish[1], Helen Li [1], Babak Behsaz[1], Olivier Elemento [2], James Zou [3], Fereydoun Hormozdiari [1,4] ✉, Babak Alipanahi [1] ✉ & Hani Goodarzi [5,6] ✉

Liquid biopsies have the potential to revolutionize cancer care through non-invasive early detection of tumors. Developing a robust liquid biopsy test requires collecting high-dimensional data from a large number of blood samples across heterogeneous groups of patients. We propose that the generative capability of variational auto-encoders enables learning a robust and generalizable signature of blood-based biomarkers. In this study, we analyze orphan non-coding RNAs (oncRNAs) from serum samples of 1050 individuals diagnosed with non-small cell lung cancer (NSCLC) at various stages, as well as sex-, age-, and BMI-matched controls. We demonstrate that our multi-task generative AI model, Orion, surpasses commonly used methods in both overall performance and generalizability to held-out datasets. Orion achieves an overall sensitivity of 94% (95% CI: 87%–98%) at 87% (95% CI: 81%–93%) specificity for cancer detection across all stages, outperforming the sensitivity of other methods on held-out validation datasets by more than ~ 30%.

Lung cancer is the leading cause of cancer mortality in the US, accounting for about 1 in 5 of all cancer deaths[1]. Each year, more people die of lung cancer than of colon, breast, and prostate cancers combined. Early detection of lung cancer improves the effectiveness of treatments and patient survival rates[2] but adherence to screening is often low[3]. Nationally, only 23% of lung cancer cases are diagnosed before metastasis (stage I–III), when the five-year survival rate is 59%.

Previous attempts for detection of lung cancer through circulating tumor DNA (ctDNA)-based tumor-informed liquid biopsy assays have a low sensitivity (55%–57%) for early-stage disease, when treatments are most effective[4,5]. While epigenomic assays have improved upon the overall sensitivity of mutation-based modalities by leveraging the cell-type specificity of DNA methylation[6,7] or DNA fragmentation patterns[8,9], sensitivity for early stage and small tumors remains low due to limited DNA shedding[10]. More recent epigenomic studies report higher sensitivities for lung cancer detection, but such a gain usually comes at the cost of the lower specificity. Mazzone et al.[11], for example, report sensitivity of 84% (95% CI 79%–88%) and specificity of 53% (95% CI: 45%–61%). Similarly, Hong et al.[12] report sensitivity of 58% (95% CI: 49%–67%) at specificity of 75% (95% CI: 71%–79%) for stage I and sensitivity of 74% (95% CI: 63%–83%) at specificity of 30% (95% CI: 24%–37%) for stage II lung cancer detection.

Reorganization of the chromatin, as commonly observed in cancer cells[13], often results in the de novo access of the cellular

[1]Exai Bio Inc., Palo Alto, CA, US. [2]Weill Cornell Medicine, New York, NY, US. [3]Stanford University, Stanford, CA, US. [4]University of California, Davis, CA, US. [5]University of California, San Francisco, CA, US. [6]Arc Institute, Palo Alto, CA, US. [7]These authors contributed equally: Mehran Karimzadeh, Amir Momen-Roknabadi, Taylor B. Cavazos. ✉e-mail: fereydounh@exai.bio; babaka@exai.bio; hani@arcinstitute.org

transcriptional machinery to previously inaccessible genomic regions[14]. Global disruptions in the RNA regulatory machinery in cancer[15] may also result in the appearance and stabilization of RNA fragments not commonly observed in normal tissues[16]. We recently reported the discovery of a class of cancer-emergent small RNA (smRNA)s, termed orphan non-coding RNA(oncRNA)s, that arise as a consequence of cancer-specific genomic reprogramming[17]. OncRNAs are abundant, stable, and actively secreted from living cancer cells into the blood[18]. We have generated a catalog of over 777,291 oncRNAs across major cancer types. Some oncRNAs, such as *T3p*, exhibit pro-metastatic roles, while others could emerge as a byproduct of repro-grammed RNA metabolism. Contrary to DNA-based assays, oncRNAs do not require cellular death to be released. Active expression and secretion of oncRNAs allows for early detection of cancer and subtype stratification in a liquid biopsy setting[18].

Since only a fraction of oncRNAs may be present in the volume of a blood draw, smRNA fingerprinting results in sparse patterns from thousands of individual oncRNAs species. Given the zero-inflated nature of oncRNA patterns, the underlying biological variation distinguishing different cancer types or separating cancer from non-cancer may become dominated by technical confounders, such as differences in sequencing depth, RNA extraction, sample processing, and other unknown sources of variation. In addition, often the sample collection process itself involves known sources of variation that should be accounted for, including biological differences between donors (age, sex, BMI, etc.). Therefore, developing a generalizable liquid biopsy assay requires effective strategies for modeling the biological properties of circulating biomarkers of interest and disentangling the technical and biological variation in sequencing data.

In recent years, various classes of neural networks have provided robust and customizable frameworks for guided representation learning. Deep generative models can leverage variational inference[19] or pre-training on masked data[20–22] to facilitate a variety of downstream tasks. Given the over-parameterized nature of these networks, a large number of samples is required for the adaptation of these models for clinical genomics applications. Furthermore, within the current framework of these models, explicit encoding of known technical variation (e.g. batch) is necessary, thus limiting the generalizability to new datasets. To overcome these challenges, we developed Orion, a two-arm semi-supervised multi-input variational auto-encoder for a liquid biopsy application using oncRNAs.

Here, we show the capability of Orion in learning a generalizable pattern of circulating oncRNAs for a variety of applications, including early detection of lung cancer, tumor subtyping, and removing batch effects in the presence of confounded signals.

## Results

The liquid biopsy and approach for cancer detection proposed here uses newly annotated lung cancer-emergent and tumor-released oncRNAs as a signature for cancer detection from blood. In this approach, using publicly available smRNA-seq data from The Cancer Genome Atlas (TCGA)[23,24], first, we discovered a set of oncRNAs; previously un-annotated scarce smRNAs that are selectively expressed in lung tumors versus normal lung tissues. Next, we used the expression of the selected oncRNA features in an in-house dataset of serum samples for cancer detection (Fig. 1a, see Methods).

We then developed a deep generative AI model, Orion, for cancer detection using the abundance of cell-free oncRNAs in serum samples (Fig. 1b). The proposed model is a generalizable approach that accounts for potential batch and vendor effects and other sources of expression variance that are not related to disease status. By removing these sources of noise, Orion improves the overall accuracy of cancer detection and is generalizable to unseen samples. At a high level, Orion uses variational inference to learn a Gaussian distribution from oncRNA data. We added several additional constraints through cross-

entropy (CE) and triplet margin loss (see Methods) to emphasize the task-relevant information (e.g. cancer vs. control) while minimizing the task-irrelevant information (e.g. differences in library size or between sample sources) within the embedding space. A cancer inference neural network then samples from this distribution to predict labels of interest including detection of cancer or tumor subtype. The model achieves these objectives by minimizing a negative log-likelihood loss based on zero-inflated negative binomial distribution to allow for the relative sparsity of biomarker measurements from the blood. We used 20% of the samples as a held-out validation dataset and the remaining samples for training within a 10-fold cross-validation setup.

### Description of datasets

**Non-small cell lung cancer (NSCLC) and tumor-adjacent normal smRNA dataset for oncRNA selection.** We used the TCGA smRNA-seq database to identify 255,393 NSCLC-specific oncRNAs through differential expression analysis of NSCLC and non-cancerous tissues (see Methods).

**smRNA data.** We generated an in-house dataset of serum collected from 1050 treatment-naive individuals (419 with NSCLC and 631 without a history of cancer). These samples are sourced from two different suppliers, where each supplier provided both cancer and control samples collected from multiple sites (Table 1, see Methods). We used 80% of these samples for model training and evaluation through 10-fold cross-validation (training dataset). During cross-validation, for each of the 10 folds, we used 90% of the samples (training set) for training 5 models with different random seeds and the remaining 10% of the samples (tuning set) for assessing the cross-validated performance of the model. We used the average score of the 50 models on the held-out 20% of the data (validation dataset). We sequenced cell-free smRNA isolated from 0.5 mL of serum to quantify the expression of NSCLC-specific oncRNAs identified in the TCGA data (Fig. 1a, see Methods). A total of 237,928 (93.15%) of the selected oncRNAs from tissue samples were detected in at least one of the samples.

### Orion model architecture

To distinguish cases from controls on the basis of their cell-free oncRNA content, we developed **Orion**; a customized, regularized, multi-input, and semi-supervised variational auto-encoder (VAE) (Fig. 1b). As a VAE, Orion uses variational Bayes objectives to learn the parameters of a zero-inflated negative binomial distribution for expression of each oncRNA. This class of distribution accounts for over-dispersion and low sensitivity which are inherent to blood-based genomic and transcriptomic measurements (Supplementary Fig. 1a–c). It has a two-arm architecture, modeling the expression of oncRNAs in one arm and the expression of annotated smRNAs in the other. The latter is used to account for differences in the size of sequencing libraries across samples. Orion also includes additional classification and contrastive learning objectives to accommodate label prediction and remove unwanted confounders in the learned representations (Fig. 1b).

The semi-supervised nature of Orion allows its representation learning to capture the biological signal of interest (e.g. cancer detection) while removing unwanted confounders (such as batch effects). The generative capability of Orion during classifier training enables learning a robust pattern of biomarkers for cancer detection. To ensure that the model learns a biologically grounded representation of the data irrespective of technical confounders, we used contrastive distance metric learning with a triplet margin loss (Fig. 1b).

To evaluate the capability of Orion in cancer detection and its generalizability, we divided our dataset into a held-out 20% and a remaining 80%. For 80% of the data, we trained Orion models in a non-overlapping 10-fold cross-validation setup. During each fold, we

identified a subset of TCGA-derived oncRNAs that within the training set, were enriched among the cancer samples compared to control samples of each data source supplier, resulting in an average of 6376 ± 60 (S.D) oncRNAs per fold. We trained 5 Orion models with different random seeds on each fold and averaged the scores on the tuning set.

Based on the cross-validated scores of the training dataset, the model achieved area under the receiver-operating characteristic curve (ROC) of 0.97 (95% CI 0.96–0.98) and overall sensitivity of 94% (95% CI 91%–96%) at 90% specificity (Fig. 2a). In an identical setup with the same set of oncRNAs for each training fold, support vector machine (SVM) classifier[25] had an area under ROC of 0.87 (95% CI 0.84–0.89) and overall sensitivity of 61% (95% CI 55%–66%). Other methods such as the commonly used ElasticNet[26] model, XGBoost[27], and $k$-nearest neighbors ($k$-NN) classifier[28] also performed worse than Orion (Supplementary Table 1). More importantly, stage I sensitivity ($n = 88$) was 90% (95% CI 83%–94%) for Orion versus 56% (95% CI 47%–65%) for the SVM classifier at 90% specificity (Fig. 2a). Sensitivity for later stages (II, III, and IV with $n = 243$) was 97% (95% CI 93%–99%) and 63% (95% CI 56%–70%) for Orion and the SVM classifier, respectively (Fig. 2b). For detecting tumors smaller than 2 cm (T1a–b, $n = 52$), Orion achieved a sensitivity of 87% (95% CI 74%–94%) at 90% specificity, while the SVM classifier had a sensitivity of 44% (95% CI 30%–59%) at 90% specificity.

In a bootstrap analysis, AUC of Orion was significantly higher than both the SVM classifier ($\Delta_{AUC} = 0.1$ (95% CI: 0.08–0.13)) and XGBoost ($\Delta_{AUC} = 0.03$ (95% CI: 0.02–0.04), Supplementary Table 1). While AUC of Orion and XGBoost were relatively similar, $F_1$ score and sensitivity of Orion at 90% specificity were also better for Orion compared to XGBoost ($\Delta_{F_1} = 0.05$ (95% CI 0.02–0.08), $\Delta_{sensitivity} = 9\%$ (95% CI 5%–13%)).

To assess the generalizability of Orion, we chose the cutoff corresponding to 90% specificity among the 10-fold cross-validated predictions, and measured various classification metrics on the held-out validation dataset. Orion demonstrated a strong agreement in performance for the held-out validation dataset, while XGBoost, ElasticNet, and other model performances were on the lower bound of their 10-fold CV measurements (Fig. 2d, Supplementary Table 1). For example, at expected specificity cutoffs of 90%, 95%, and 99%, based on the cross-validated model scores, Orion had observed specificity values of 87% (95% CI: 80%–93%), 94% (95% CI: 89%–98%), and 98% (95% CI: 94%–100%), while other methods had inconsistent ranges of specificity (Supplementary Table 2).

As a measure of successful batch effect removal, we expected the model scores for control samples to be similar, and therefore, not distinguish the sample suppliers. Orion had an area under ROC of 0.53 (95% CI 0.47–0.58), suggesting it successfully removed the impact of

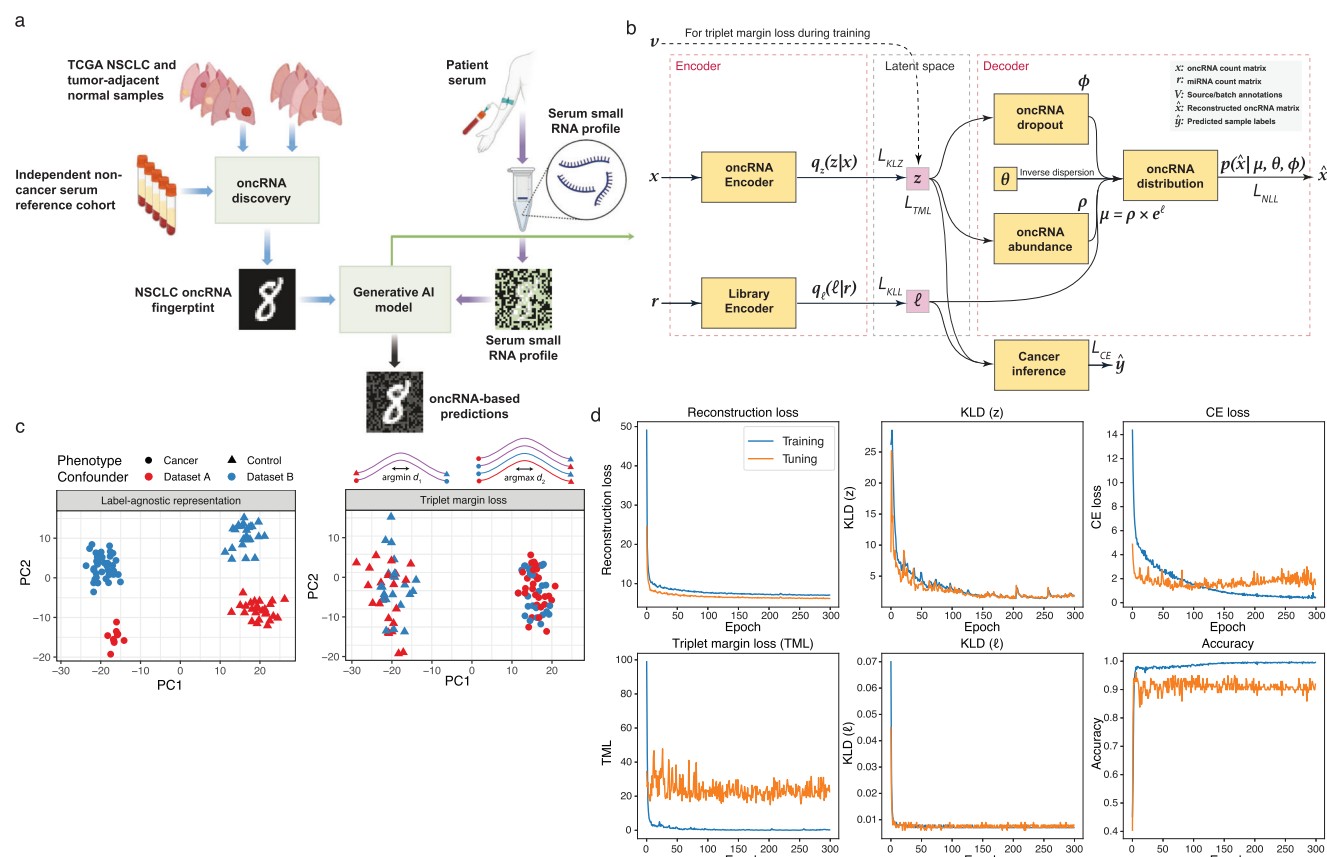

**Fig. 1 | oncRNA-based liquid biopsy platform and Orion architecture. a** We discovered NSCLC oncRNAs from TCGA tissue datasets and investigated them in the blood of patients with NSCLC and non-cancer controls. We showed an analogy depicting NSCLC oncRNA fingerprint as a hand-written digit, serum oncRNA fingerprint as a noisy pattern, and generative AI embeddings as a denoised version. Created in BioRender. Alipanahi, B. (2024) BioRender.com/b61n795. **b** Orion architecture requires two input count matrices for oncRNAs ($x$) and endogenous expressed RNAs ($r$). Each input is fed to a standard VAE where the objective is to learn a joint representation of oncRNA counts under a zero-inflated negative binomial distribution (right). A joint embedding will be used by the cancer inference neural network for classification tasks (bottom right). **c** Schematic of triplet margin loss application on simulated data. The left panel shows a label-agnostic embedding, and the right panel shows an embedding with a triplet margin loss constraint to minimize technical variations while preserving biological differences. For each sample, we use positive anchors (same phenotype, different dataset) and negative anchors (different phenotype, any dataset) to minimize or maximize the embedding distance, respectively. **d** Loss convergence plots show convergence of 5 of the losses of Orion as well as classification accuracy during training.

**Table 1 | Sample demographics**

| Demographics | | Training dataset | | Validation dataset | |
|---|---|---|---|---|---|
| | | Control | Cancer | Control | Cancer |
| Sample size | Count, $n$ | 506 | 334 | 125 | 85 |
| Age | Mean (SD) | 62.18 (11.75) | 65.84 (9.60) | 61.80 (10.80) | 63.85 (10.35) |
| Sex | Female (%) | 238 (47.04%) | 125 (37.43%) | 50 (40.00%) | 40 (47.06%) |
| Smoking status | Never-Smoked, $n$ (%) | 271 (53.56%) | 34 (10.18%) | 71 (56.80%) | 7 (8.24%) |
| BMI | BMI obese (≥ 30), $n$ (%) | 124 (24.51%) | 72 (21.56%) | 28 (22.40%) | 15 (17.65%) |
| Race | White, $n$ (%) | 253 (50.00%) | 220 (65.87%) | 62 (49.60%) | 55 (64.71%) |
| | Black/African American, $n$ (%) | 54 (10.67%) | 12 (3.59%) | 14 (11.20%) | 1 (1.18%) |
| | Asian, $n$ (%) | 15 (2.96%) | 4 (1.20%) | 3 (2.40%) | 0 (0.00%) |
| | Other/Unknown, $n$ (%) | 184 (36.36%) | 98 (29.34%) | 46 (36.80%) | 29 (34.12%) |
| Ethnicity | Hispanic, $n$ (%) | 179 (35.38%) | 12 (3.59%) | 46 (36.80%) | 5 (5.88%) |
| | Non-hispanic, $n$ (%) | 281 (55.53%) | 316 (94.61%) | 59 (47.20%) | 80 (94.12%) |
| | Other/Unknown, $n$ (%) | 45 (8.89%) | 6 (1.80%) | 19 (15.20%) | 0 (0.00%) |
| Source | Indivumed, $n$ (%) | 183 (36.17%) | 258 (77.25%) | 46 (36.80%) | 65 (76.47%) |
| | MT Group, $n$ (%) | 323 (63.83%) | 76 (22.75%) | 79 (63.20%) | 20 (23.53%) |

Sample size and key demographic aspects of training dataset and held-out validation dataset.

suppliers, while XGBoost and SVM classifier had higher area under ROCs of 0.59 (95% CI 0.54–0.64) and 0.57 (95% CI 0.52–0.62), respectively.

Given that the control samples in our cohort had an over-representation of individuals without smoking history compared to the cancer samples (54% vs. 10%), we examined the impact of smoking status of samples on model scores. We found that among control samples, Orion validation set score had an area under ROC of 0.6 (95% CI 0.5–0.7) with respect to presence of smoking history, further confirming little variation of the model score for individuals with or without a history of smoking.

To identify the most important oncRNAs for the model, we used SHapley Additive exPlanations (SHAP)[29] average values among model folds. Among the high-SHAP oncRNAs for the model, we observed overlap or vicinity of oncRNAs to some of the genes with significance in lung cancer etiology and prognosis. These included *SOX2-OT*[30], *HSP90AA1*,[31,32] and *FZD2*[33] (Fig. 2e).

To understand the model architecture components of Orion contributing most to high performance and limited batch detection, we performed a series of ablation experiments. We trained multiple models which lacked one or more of Orion's features, such as triplet margin loss, cross entropy loss, reconstruction loss, or generative sampling for computation of the cross entropy loss during training. We found that triplet margin loss allows the model to minimize the impact of the technical variations (Fig. 3a). Generative sampling allows the model to achieve higher overall performance and better cross-entropy loss convergence (Fig. 3b). Orion's embeddings in the presence of all of its components, particularly with triplet margin loss and generative sampling, result in a better separation of cancer samples from control samples, which allows Orion's classifier to operate on a representation of the data with minimal technical variations (Fig. 3c). The presence of different components of Orion, particularly the reconstruction loss, result in a better convergence of the test-set cross entropy loss (Fig. 3d).

We hypothesized that training the classifier of the model by sampling from the learned distribution allows Orion to achieve higher robustness and performance at a smaller sample size. In comparison with an identical architecture where the classifier uses the expected value of the distribution instead of sampling, we observed a significant improvement in convergence and generalizability of the cross entropy loss with smaller sample sizes (Fig. 3b–d).

To assess if Orion learns more informative task-relevant embeddings than commonly used methods such as principal component analysis (PCA)[34] or Harmony[35], we examined how these embeddings compare in downstream tasks. We provided Harmony with the same variables for batch correction as Orion's triplet margin loss (sample supplier and experiment ID). While Orion's key clusters reflect cancer and control labels (Supplementary Fig. 2, projected here in UMAP space solely for visualization), the naive representation of Harmony and PCA fail to capture this key biological variability. Next, we trained an XGBoost model on the training set and evaluated the performance in cancer detection from the embeddings in the tuning set. Label-agnostic batch correction of Harmony resulted in loss of biological information and a worse performance than PCA, while Orion out-performed both PCA and Harmony with at least 30% higher sensitivity at 90% specificity (Supplementary Fig. 2).

Given that Orion uses raw count of oncRNAs as one of the inputs and normalization occurs internally, next we investigated Orion scores on an in-house dataset with pool of 3 control samples, divided into 7 technical replicates, and sequenced at 5 different target depths ranging from 4 to 60 million reads. These samples were predicted as controls among all sequencing depths and correlation of the scores with sequencing depth was minimal (linear model adjusted $R^2$ of 0.154 (95% CI 0.004–0.546), Supplementary Fig. 3a–b).

To better understand the sensitivity of Orion for detection of cancer samples, we combined the sequencing reads from cancer and control samples at different ratios. We noticed that Orion cancer calls from the validation set can tolerate up to 40% of dilution without an impact on sensitivity, a property that we did not observe for other methods (Supplementary Fig. 3c).

To establish if Orion model scores are impacted by a small number of oncRNAs or they can leverage a large number of oncRNAs, we investigated how in silico perturbation of oncRNAs impact model predictions (Supplementary Fig. 4). We perturbed the top-SHAP oncRNAs, the oncRNAs with the highest SHAP scores as measure of their importance for the model, in two different ways. In an ablation experiment, we set such oncRNA counts to zero among all samples of the validation set. This approach resulted in a decrease in scores of the cancer samples, impacting sensitivity and area under ROC particularly when ablating more than 1000 top-SHAP oncRNAs. In another experiment, we permuted the values of top-SHAP oncRNAs among all cancer and control samples, resulting in control samples to artificially

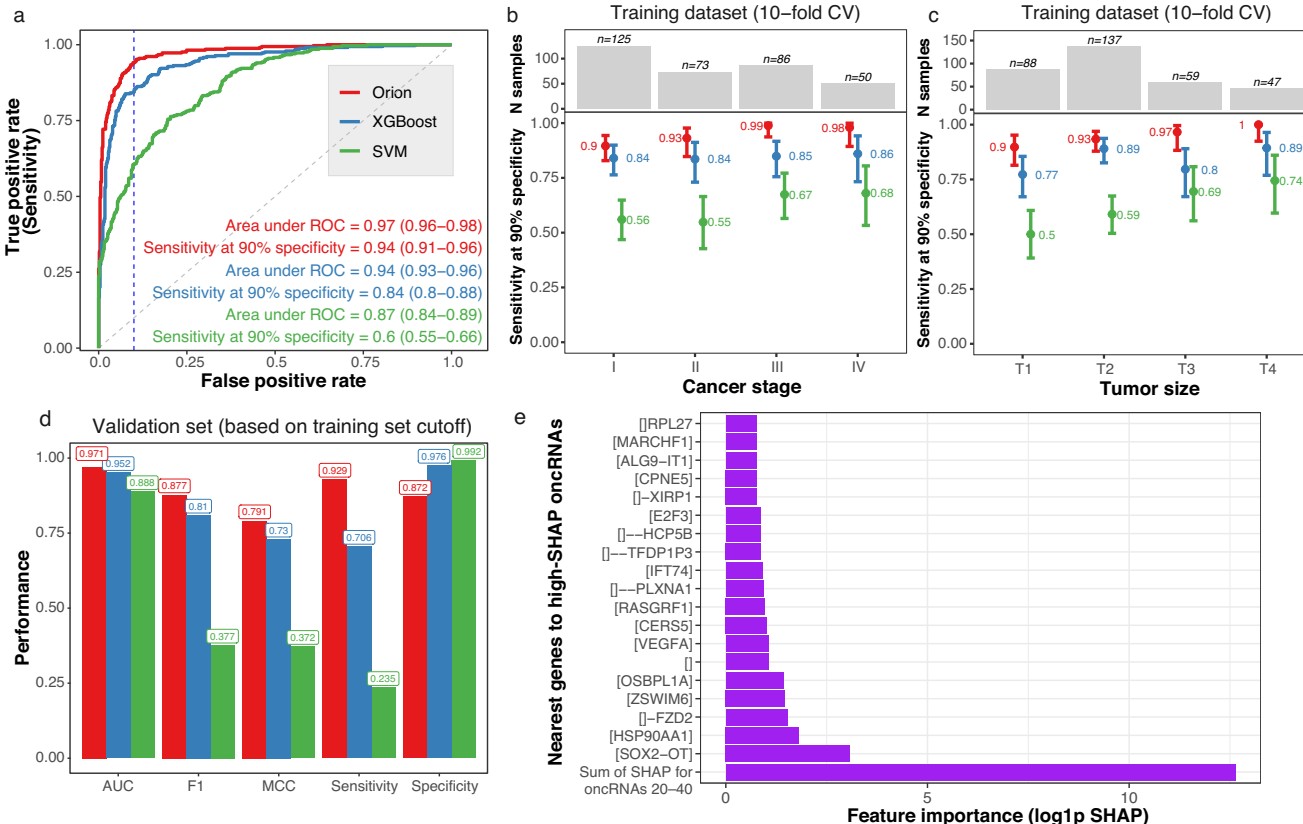

**Fig. 2 | Model performance on training and validation set. a** The ROC plot on the tuning set of 10 non-overlapping folds of model training for Orion (red), XGBoost (blue), and SVM classifier (green). The vertical blue line shows specificity at 90%. The text shows the area under ROC and sensitivity at 90% specificity with 95% confidence intervals. **b** Sensitivity of the model for tumors of different cancer stages at 90% specificity for Orion (red), XGBoost (blue), and SVM classifier (green). Error bars indicate the 95% confidence interval. The bar plot shows the number of samples in each category. **c** Sensitivity of the model stratified by T score (size) similar to (**b**). **d** Performance measures of binary classification in the held-out validation dataset. We computed all threshold-dependent metrics (all except area under ROC) based on the cutoff resulting in 90% specificity in the 10-fold cross validated training dataset. The bar height shows the point estimate of area under ROC, $F_1$ score, Matthew's correlation coefficient (MCC), sensitivity, and specificity. **e** Barplot shows log1p of SHAP score (x-axis) for the top 20 oncRNAs (y-axis). Y-axis labels indicate the nearest gene to the oncRNA. The first rows shows the sum of the next 20 oncRNAs (oncRNAs ranked 21st to 40th by their SHAP score). For gene A, [A] indicates overlap, []A indicates 1 kbp distance, [] − A indicates 10 kbp distance, [] − − A indicates 100 kbp, and [] indicates no genes within 1 Mbp distance.

express cancer features. This approach resulted in a noticeable increase in model scores of control samples and decrease in specificity. Overall, the method behaved as expected from a cancer detection perspective, showing robustness when top-SHAP cancer-oncRNAs are absent, while showing high sensitivity to expression of top-SHAP cancer-oncRNAs in samples that lack most cancer-oncRNAs.

## Orion can identify tumor subtype from circulating oncRNAs
In addition to the early detection of cancer signals in patients with NSCLC, understanding tumor histology has major implications in therapy selection and resistance mechanisms. Squamous cell carcinoma transformation of lung adenocarcinoma has been reported to take place spontaneously[36] or after targeted therapy resistance. Such mechanisms of acquired resistance have been reported for epidermal growth factor receptor (EGFR) inhibitors, tyrosine kinase inhibitors (TKIs)[37], KRAS inhibitors[38], and immunotherapies[39]. Traditional methods of stratifying patients to evaluate for squamous cell carcinoma transformation involve repeat biopsies of a lung cancer patient which can lead to severe side effects such as pneumothorax, hemorrhage, and air embolism[40].

We had previously observed that given the tissue-specific landscape of chromatin accessibility in different cancers, oncRNA expression patterns are unique to cancer types and subtypes[41]. We hypothesized that biological differences of lung adenocarcinoma and squamous cell carcinoma would also be reflected in cell-free oncRNA

content, allowing us to distinguish these major subtypes of NSCLC. While tumor tissues are vastly different from normal tissue, the differences in subtypes of a given tumor are far less substantial. In NSCLC, for example, the agreement of pathologists for different subtypes is approximated to be 0.81[42]. As a result, tumor histology subtype prediction is more difficult than cancer detection.

To evaluate our hypothesis, we investigated the potential of distinguishing two major NSCLC subtypes, adenocarcinoma and squamous cell carcinoma, using oncRNAs in blood. For this analysis, we used 20-fold cross-validation to adjust for the reduction in the number of samples given that this is a NSCLC-specific task. For later stage tumors (stages III/IV), Orion achieved an area under ROC of 0.75 (95% CI: 0.67–0.83) and a sensitivity of 71% (95% CI: 56%–84%) at 70% specificity in distinguishing squamous cell carcinoma from adenocarcinoma samples in serum samples (Fig. 4).

## Discussion
Variational inference serves as the backbone of a plethora of deep generative models, particularly for single-cell genomics applications[19]. The flexibility of these models allows for reference building through transfer learning[43] or modeling specific perturbations through contrastive learning[44]. However, when biological signals are weak or scarce, as is the case in liquid biopsies where we are in search of a needle in the haystack, technical confounders that are due to differences in sequencing platforms or data sources become more

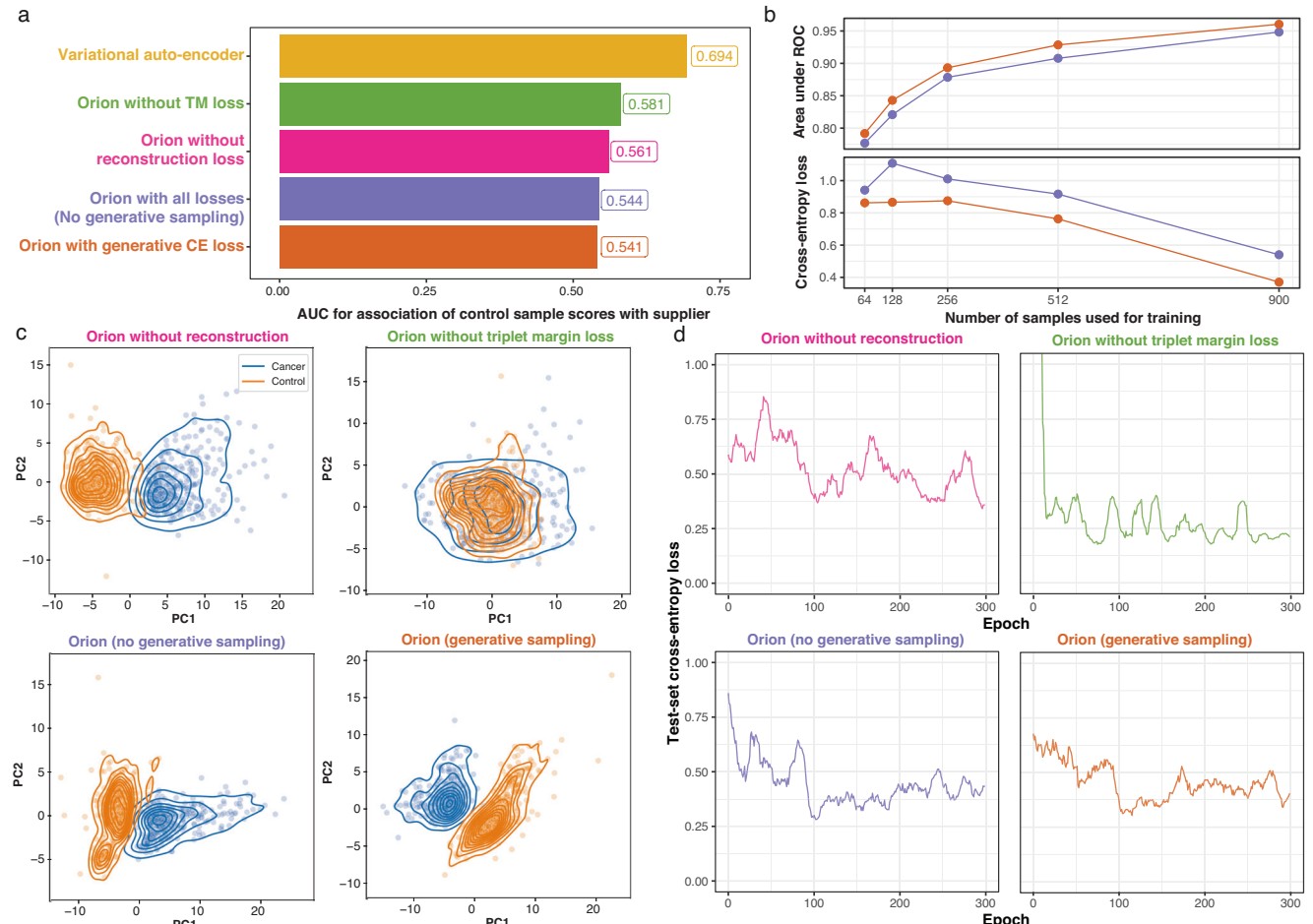

**Fig. 3 | Ablation of Orion components. a** Area under the ROC of 5 different models when comparing score of the control samples with respect to the sample supplier. **b** Area under ROC (top panel) and cross entropy loss (bottom panel) for cancer detection as a function of the number of samples used during training. Orange shows Orion with generative sampling for computation of cross-entropy loss during training, and purple shows Orion without this feature. **c** Scatter plots overlaid with kernel density estimates show cancer (blue) and control (orange) samples based on the first two principal components of Orion's embedding space in 4 different conditions. **d** Test-set cross entropy loss of the same models.

pronounced. As a result, without any intervention, the naive representation learning may regress out the signal of interest, as was the case with PCA and even the state-of-the art batch correction method, Harmony[45] (Supplementary Fig. 2). Representation learning, therefore, is rarely used for clinical genomics applications. Instead, classical regularized supervised learning methods (e.g., ElasticNet) are adopted, which are able to resolve the $p$ (number of features) $>> n$ (number of samples) problem by finding an adequate balance between the number of features the model utilizes and the individual weight of each feature. While these methods have been extensively applied in clinical genomics and liquid biopsy, they fail to model non-linear interactions among the input features and the higher-order patterns in the data.

Here we sought to leverage representation learning for obtaining an abstract low-dimensional embedding of cell-free oncRNAs. We hypothesized that a deep generative AI model can augment the downstream classifier to learn robust and generalizable patterns of cancer-specific oncRNAs. This approach not only reduces the number of features by approximately 300 fold, but it can also enhance the number of unique samples the classifier is trained on through generative sampling, essentially converting $p \gg n$ to a favorable $n \gg p$. A key aspect to the success of our approach is tailoring the process of representation learning through the addition of contrastive learning[46] (Fig. 3). Inherently, these objectives are in contradiction, one enforcing the latent distribution to preserve all sources of data variation, while the other imposes a constraint to remove unwanted variations. As a result, these two objectives meet at the balancing minima of a sacrifice in reconstruction at the gain of emphasizing the biological differences among the samples.

Here we demonstrated the success of our approach in training a model that not only achieved superior performance for cancer detection, but also exhibited generalizability to held-out datasets. As a machine learning method, however, we do not expect Orion to generalize to out-of-distribution samples. In fact, real-life applications of machine learning models must include detection of out-of-distribution samples to avoid generating spurious predictions[47]. The deviation of Orion loss terms for new samples has the potential of facilitating the identification of out of distribution samples.

Contrary to other methods, Orion scores remained unchanged among samples coming from different sources or with different smoking histories, underscoring the robustness of our model. Orion demonstrates promising performance in predicting tumor subtypes from blood, even with the challenges posed by the lack of clear ground truths in histopathological calls. Given that the pathologist agreement for this task is itself around 80%[42] and the observation that our model improved by increasing the number of samples in the training set (Fig. 3b), a larger dataset with molecularly-assigned labels could provide an opportunity for liquid histology applications beyond cancer detection using Orion.

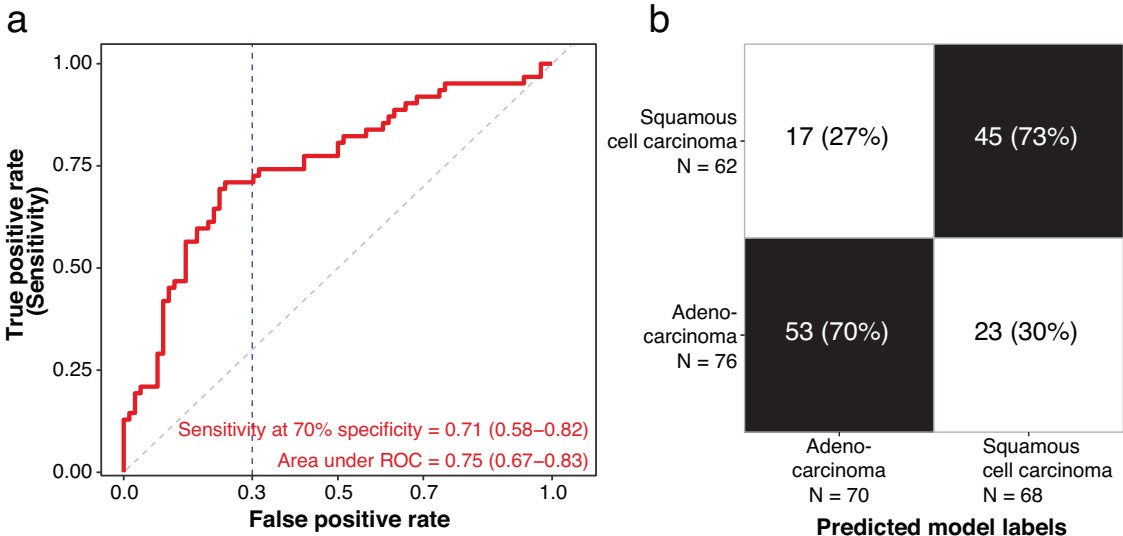

**Fig. 4 | Orion allows distinguishing tumor subtypes from the oncRNA profiles of the blood. a** ROC plot of Orion for distinguishing squamous cell carcinoma from adenocarcinoma among stage III/IV NSCLC samples. **b** Confusion matrix of Orion's subtype prediction at 70% specificity cutoff.

While the adaptation of deep learning models in clinical genomics is in its early days, our results establish a strong case for the potential of generative AI in advancing the applications of liquid biopsy, as well as liquid histology. The potential of liquid histology in monitoring tumor subtype transitions, for example, may allow for more patients to benefit from targeted therapy of emerging tumor populations. The combination of our liquid biopsy platform for profiling a stable, abundant, and cancer-specific biomarker−oncRNAs−and our generative AI model which is compatible with blood-based measurements, provides a novel opportunity for filling a clinical gap in sensitive and early cancer detection and monitoring.

## Methods

### Dataset

Here, we used an in-house dataset of serum collected from 1050 treatment-naive individuals sourced from two different suppliers: Indivumed (Hamburg, Germany; 229 controls and 323 NSCLC cases) and MT Group (Los Angeles, CA; 402 controls and 96 NSCLC cases). Each supplier also collects samples from multiple sites. Previous studies on early detection of lung cancer through liquid biopsy have sample sizes ranging from 288 to 799[7–9]. Our objective was to generate more samples than existing studies to allow us to explore the potential of deep generative models in liquid biopsy applications. We collected the entirety of the samples that we could acquire from our sources, while ensuring good representation of samples across stages of the disease. The dataset included 157 stage I, 93 stage II, 106 stage III, and 63 stage IV NSCLC cases. We used RNA isolated from 0.5 mL of serum from each donor to generate and sequence smRNA libraries of each sample at an average depth of $19.8 \pm 5.8$ million 50 bp single-end reads. The NSCLC samples included 222 samples with adenocarcinoma, 160 samples with squamous cell carcinoma, and 37 samples with unknown histological type (Table 1). Despite the challenges of collecting samples from healthy seniors without smoking history, NSCLC and control arms included both smoker and non-smoker samples and similar distribution with respect to age, sex, and body mass index (BMI). Given the imbalance of individuals with smoking history among cases and controls, we observed that the Orion model score did not vary as a function of smoking history among control samples.

### Sample exclusion criteria

During sample selection, we excluded samples of individuals matching any of the following criteria:

- Cancer patient is not treatment-naive at time of collection
- Age ≤ 18
- Prior cancer diagnosis (with or without treatment)
- Any surgery within one month of collection
- Received non-cancer systemic immune modulation therapy within 60 days of collection (e.g. monoclonal antibodies)
- History of organ transplantation
- Infusion of any blood products within 30 days of collection
- Current active COVID-19 infection
- Any prior history of cancer therapy (e.g., surgical, radiation, or medical including neoadjuvant treatment)
- Current or prior pregnancy within 12 months of collection

### Laboratory workflow

**Sample Acquisition and initial processing.** Serum samples were acquired from vendors Indivumed (Hamburg, Germany) and MT Group (Los Angeles, CA). All samples were acquired under valid IRB approvals. Serum samples were collected according to standard protocols and frozen at −80 °C after processing.

**RNA extraction and library preparation.** RNA was extracted from 1 mL of serum either using the Quick-cfRNA Serum & Plasma Kit (Zymo Research, Cat # R1059). Libraries were prepared using the SMARTer smRNA-Seq Kit for Illumina (Takara, Cat # 635031) using custom RT primers to incorporate UMIs. Libraries were PCR amplified using Takara SeqAmp DNA Polymerase (Cat # 638504) and indexed during the PCR reaction. Amplified libraries were combined and size selected using either 8% NuPAGE TBE gels (Thermo Fisher, Cat # EC6215BOX) or BluePippin (Sage Science, Cat # BDQ3010) using 3% Agarose Cassette DF Marker Q3 (Sage Science). Pooled libraries were purified using the Nucleospin Gel and PCR Cleanup Kit (Machery Nagel, Cat # 740609.50). Purified libraries were quantified either using the Tapestation using High Sensitivity D1000 Sample Buffer (Agilent, Cat # 5067-5585) or by using the qPCR-based NEBNext library quant kit for Illumina (NEB, Cat # E7630) and sequenced on a NextSeq 2000 instrument (Illumina) using P3 flowcells, 50 bp single-end, to an average depth of $19.8 \pm 5.8$ million mapped reads.

## Data processing

For processing of raw sequencing files, we used bclconvert (v4.0.3), cutadapt[48] (v4.1), FASTQC (v0.12.1), UMICollapse[49] (v1.0.0), Bowtie-2[50] (v2.4.5), samtools[51] (v1.16.1), pysam[52] (v0.20.0), and bedtools[53] (v2.30.0).

## Orion architecture

Orion is a variational auto-encoder[54], adapting scVI[19] with additional input, connections, and objectives for removing known sources of technical variation as well as performing regression or classification tasks. Let $x_i \in \mathbb{Z}_+^d$ and $r_i \in \mathbb{Z}_+^m$ denote counts for $d$ oncRNAs and $m$ endogenous highly-expressed smRNAs for the $i$-th sample, respectively. Moreover, let $y_i \in \{0,1\}^b \times \mathbb{R}^t$ and $v_i \in \mathbb{Z}_+^c$ denote the $b$ binary and $t$ real targets (cancer status) and the $c$ known confounders (sample source, processing batch, etc), respectively.

The core idea is that there are linear and nonlinear dependencies between different oncRNAs, e.g., they are generated due to disruption in the same pathway hence their counts are correlated. Therefore, we will be able to project the space of $\mathcal{X}$ − that can be very high-dimensional − onto a low-dimensional latent space $\mathcal{Z}$ using a mapping $f_z : \mathcal{X} \to \mathcal{Z}$ (called oncRNA encoder), while capturing the essence of variation in $\mathcal{X}$. This means that we could find a mapping $g : \mathcal{Z} \to \mathcal{X}$ (called decoder), such that $\hat{x} = g(z) = g(f(x))$ is approximately the same as $x$, e.g., $||x - \hat{x}||_2^2$ is small. In variational auto-encoders instead of deterministically mapping $x$ to $z$, we map $x$ to a (usually Gaussian) distribution $q_z(z|x)$. When reconstructing $x$, we sample from $z \sim q_z(z|x)$ and using this sample, we can generate a distribution for reconstructed $x$ as $\hat{x} = p_x(x|z)$.

A common source of variation in transcriptomic data originates from the total sequenced RNA. An oncRNA might not be observed for two reasons: either it does not exist and is not secreted or it is indeed in blood but due to low-volume blood sampling or limited sequencing, it has not been picked up in the experiment. We assume that $z$ will take care of the former, but for the latter effect, $\ell \in \mathbb{R}$ is another unobserved random variable that accounts for input RNA level and library sequencing depth. Here, since oncRNA counts $x$ are usually small and unsuitable for computing library size − unlike scVI − we use a set of endogenous highly-expressed RNAs and an additional encoder $f_\ell : \mathcal{R} \to \ell$ to compute a normal distribution $q_\ell(\ell|r)$ as a proxy for the log of library size. In other words, the library size is log-normal with priors originating from the log of mean and variance of $\sum_m r_i$ in a given mini-batch. As a result, $\ell$ shows a strong correlation with the total number of oncRNA reads, even though it is not derived from oncRNAs (Supplementary Fig. 1a, b).

Similar to gene counts across cells in single-cell RNA-seq data, any oncRNA is observed in only a few samples and its counts are mostly zeros, also called zero-inflated. We assume the non-zero counts follow a negative binomial distribution. Inspired by scVI[19], we model the oncRNAs count as a conditional zero-inflated negative binomial (ZINB) distribution $p(x|z, \ell)$, where $z \in \mathbb{R}^k$, $k \ll d$ is the latent embedding of $x$.

Orion decoders learn the zero-inflation parameter $\phi_i$ through $f_\phi : \mathcal{Z} \to \phi$ and the transcription scale parameter $\rho_i$ through $f_\rho : \mathcal{Z} \to \rho$. $f_\rho$ involves a softmax step, enforcing representation of the expression of each oncRNA as a fraction of all expressed oncRNAs.

In the Gamma-Poisson representation of the negative binomial distribution, $\mu = \rho_i \times e^{\ell_i}$ will provide the shape parameter of the Gamma distribution, and input-independent learnable parameter $\theta$ will represent the inverse dispersion.

In short, to train Orion:

1. We learn a low-dimensional Gaussian distributions $q_z(z|x)$ and $q_\ell(\ell|r)$, so that zero-inflated negative binomial distribution $q_x(x|z, \ell)$ has the generative capability of producing realistic in silico oncRNA profiles. To do so:

(a)  We minimize

$$L_{KLZ} = D_{KL}(q_z(z|x)||p(z)), \tag{1}$$

where $D_{KL}$ is the Kullback-Leibler divergence[55] and $p(z) = \mathcal{N}(\mathbf{0}, I)$ is the prior distribution for $z$.

(b)  We minimize

$$L_{KLL} = D_{KL}(q_\ell(\ell|r)||p(\ell|r)), \tag{2}$$

where $p(\ell|r)$ is the prior log-normal distribution for $\ell$. Unlike $z$, the prior distribution for $\ell$ is different from batch to batch and its log-mean and log-standard deviation are computed based on values of $r$ in each mini-batch $\mathcal{B}$.

2. We minimize the reconstruction loss by minimizing the negative log-likelihood of a zero-inflated negative binomial distribution describing the distribution of the input oncRNA data:

$$L_{NLL} = -\sum_i \log p_x(x_i|\mu_i, \theta_i, \phi_i), \tag{3}$$

where $\mu_i$ is the product of the softmax of $f_\rho$ (representing transcription scale of each oncRNA) and $e^{\ell_i}$; and $\theta_i$, $\phi_i$ represent inverse dispersion and zero-inflation probability[19], respectively (Fig. 1b).

3. We use contrastive learning (triplet margin loss) to minimize the impact of known confounders $v$ on $z$. For example, this ensures that all the cancer samples from different sources are projected in proximity of each other (see Triplet Margin Loss section).

(a)  Minimize the distance between samples that have the same label (e.g. all cancer samples or all control samples) but are from a different confounder group (e.g. source, supplier, etc.) in the oncRNA embedding space $z$

(b)  Maximize the distance between samples that have different labels.

$$L_{TML} = \frac{1}{w \times c} \sum_{i \in \mathcal{B}} \sum_{(i,j,j') \in \mathcal{T}_i} \max(||z_i - z_j||_2^2 - ||z_i - z_{j'}||_2^2 + \alpha, 0), \tag{4}$$

4. We use supervised learning such that the low-dimensional embeddings $z$ are used for regression (smooth $L_1$-loss[56]). For classification, we minimized the cross-entropy loss $L_{CE}$ to predict the provided sample labels during training (e.g. cancer vs. control)

We minimize the summation of these 5 losses with weights as hyperparameters:

$$L_{Orion} = \lambda_1 L_{KLZ} + \lambda_2 L_{KLL} + \lambda_3 L_{NLL} + \lambda_4 L_{TML} + \lambda_5 L_{CE} \tag{5}$$

**Triplet margin loss.** For each sample $i$, we sample $\omega$ triplets for each confounder $v_i^c$ as follows:

1. Randomly pick a "positive" anchor $j \neq i$ such that they share the same classification label $y_i = y_j$, but do not share the same confounder $v_i^c \neq v_j^c$.
2. Randomly pick a "negative" anchor $j' \neq i$ such that they do not share the same classification label $y_i \neq y_{j'}$.
3. Add $(i, j, j')$ to $\mathcal{T}_i$, the set of triplets for $i$.

At the end of this process, each sample will have $|\mathcal{T}_i| = \omega \times c$ triplets picked for it, where $\omega$ is a hyperparameter set to 16.

During training we add a cost function that moves samples from different sources or processing batches that share the same label (e.g.,

cancer samples from different sources) closer to each other, while moving samples with different labels (e.g., cancer samples from non-cancer samples) further apart:

$$L_{TML} = \frac{1}{w \times c} \sum_i \sum_{(i,j,j') \in \mathcal{T}_i} \max(||\boldsymbol{z}_i - \boldsymbol{z}_j||_2^2 - ||\boldsymbol{z}_i - \boldsymbol{z}_{j'}||_2^2 + \alpha, 0), \quad (6)$$

where $\alpha$ is a hyperparameter that enforces what should be the minimum difference of distances between a sample and its positive and negative anchors in the latent space, and it is set to $\alpha = 1$.

## Model parameters

On its default mode used in this study, Orion has 1 hidden layer for encoding oncRNAs with 1500 hidden units, 1 hidden layer for encoding library size from endogenous RNAs with 1500 unit, an embedding space of $d = 50$ latent variables for learning the Gaussian distribution underlying the oncRNA data, an embedding space of $s = 1$ latent variable for learning the library size distribution from endogenous RNAs, and one hidden layer for decoding oncRNA data from the latent distribution. We used dropout ($p = 0.5$), $L_2$ regularization ($L_2 = 2$). The classification layer has 1 hidden layer of size 25, mapping the 50 normalized latent values to generative predictions for each class.

Orion encoders have a hidden layer of size 1500 and map $X$ to parameters of $\boldsymbol{z}_d$ with 50 dimensions and map $Q$ to parameters of $\boldsymbol{z}_s$ with 1 dimension.

The model performs classification through a 2-layer perceptron head. The input of the classification head comes from the batch-normalized product of oncRNAs and library size embeddings, i.e., $\boldsymbol{z} \times \ell$. during training, we sample from $q_z(\boldsymbol{z}|\boldsymbol{x})$ $\eta = 100$ times for each data point to improve model robustness and sensitivity to noise. At test time, we use the deterministic expected values of $\boldsymbol{z}$ and $\ell$.

## Identifying oncRNAs

To identify a set of orphan non-coding RNAs, we utilized smRNA-sequencing data from 10,403 tumor and 679 adjacent normal tissue samples from TCGA spanning 32 unique tissue types. Quality control was applied to the GRCh38-aligned BAM files to remove reads that were <15 base pairs or were considered low complexity based on a DUST score > 2[57]. Additionally, we removed reads that mapped to chrUn, chrMT, or other non-human transcripts. After filtering, we identified de novo smRNA loci by merging all reads across the 11,082 TCGA samples and performing peak calling on the genomic coverage to identify a set of smRNA loci that were < 200 base pairs. This resulted in 74 million distinct (chromosome:start–end:strand), non-overlapping candidate loci having at least one read mapping to a unique position in the genome in at least one sample from 10,403 total number of tumor samples.

For discovery of lung tumor-specific oncRNAs, we restricted to lung tumors ($n = 999$) and all adjacent normals ($n = 679$) and filtered the candidate loci for those that appeared in at least 1% of samples resulting in 1,293,892 smRNAs. We then used a generalized linear regression model to identify those smRNAs that were significantly more abundant in lung tumors compared to normal tissues. Our model adjusted for age, sex, and principal components to capture the global smRNA expression variability across tissues and batches. After multi-testing correction we restricted to suggestively significant smRNA features (FDR $q < 0.1$) that were enriched in lung tumors (OR >1) resulting in ~260k lung-tumor associated oncRNAs for downstream applications in serum.

## Training and evaluation strategy

Our dataset included a total of 1257 samples obtained from 1050 patients, with 189 samples having been sequenced more than once. We used 20% of the patients as a held-out validation dataset, ensuring an equal representation of suppliers, histological subtype

(adenocarcinoma and squamous cell carcinoma), and patient cohort (NSCLC or control) among the training and held-out validation datasets. A subset of the 189 samples with multiple libraries were randomly assigned to the training set. This resulted in 314 samples obtained from 150 patients where each of these patients had at least 2 replicates. During training, all replicates of each sample were allocated either into training or tuning set during cross-validation to avoid data leakage. Only one of the replicates of each sample was used for reporting the final model performance.

Within the training set, we used a similarly stratified 10-fold cross-validation to select the oncRNAs and train the model on the training set. Each data split ensured samples of the same patient were either in the training or test splits. We reported the performance measures only for one sample of each patient. We train 5 models per fold, each trained with a different random seed. The score of the tuning set of each fold was averaged over these 5 models. The training set performance measures are based on the held-out set of each fold. For the held-out validation dataset, we use the average of the 50 models (5 models for each of the 10 folds). We defined the model cutoff based on the cross-validated scores of the training set and reported the performance for the held-out validation dataset using that cutoff.

## Feature selection

We used the The Cancer Genome Atlas (TCGA) smRNA-seq database to identify 255,393 NSCLC-specific oncRNAs. Each tissue sample expressed a mean of 37,115 ± 14,457 S.D. of these oncRNAs. After processing serum samples for the present study, 237,928 (93.16%) of these oncRNAs were detected in at least one sample.

Within each fold of the training set, we identified oncRNAs present in at least 2% of the training set samples provided by each supplier. Additionally, for training set samples of each supplier, we identified oncRNAs that were over-represented in the cancer samples (log odds ratio > 0). Within each training fold, we selected oncRNAs passing these criteria in both of the suppliers (MT Group and Indivumed). Among the features passing these criteria, we performed 8 rounds of XGBoost classification within the training set, each time setting aside oncRNAs with non-zero Gini impurity index as a measure of feature importance. This resulted in obtaining an average of 6376 ± 60 oncRNAs in each model fold and a total of 14,014 oncRNAs identified in at least 1 fold.

## Benchmarks

**Training other models.** We used normalized oncRNA counts by dividing $\boldsymbol{x}_i$ by the total number of highly-expressed small RNA reads $\boldsymbol{r}_i$ as a surrogate of the the sequencing library depth:

$$\frac{1000 \times \boldsymbol{x}_i}{\sum_{m=1}^{m} \boldsymbol{r}_{i,m}}, \quad (7)$$

where $\boldsymbol{r}_{i,m}$ is the counts of $m$-th smRNA for sample $i$. We used scikit-learn's `StandardScaler` on the training set of each fold, and applied it on test-set or held-out set for utilizing the model. We used scikit-learn's `LogisticRegressionCV` to identify the best set of hyperparameters in a 2-fold cross-validation setup within the training set. The hyperparameters included $L_1$ ratios `[0, .1, .5, .7, .9, .95, .99, 1]` and the default $C$ parameters. The best hyperparameters were provided to a scikit-learn `LogisticRegression` model for training on the entire training set. ElasticNet models used identical oncRNAs and samples as Orion. For other models including XGBoost, SVM classifier, and $k$-NN, we used the default parameters.

**Embedding benchmarks.** In this study, Orion has an embedding space with a multi-variable Gaussian with a dimension of 50. We used 50 principal components from the same oncRNAs (scaled to total miRNA content). We fed the PCA matrix to harmony, specifying sample source

and experiment ID as `batch_key` parameter. These are the same variables that we used to guide triplet margin loss.

We used Orion's embeddings from the training set and the same subset of PCA and harmony for training XGBoost models to predict cancer with default parameters. We applied the model on Orion's embeddings from the test as well as the PCA and harmony for the same subset of samples (Supplementary Fig. 2).

**Dilution assay.** We performed an in silico dilution experiment to evaluate the performance of the different models when reads from a cancer sample are mixed in silico with reads from a non-cancer sample, at different mixture ratios. From the validation set samples, we selected 10 highest-scoring cancer samples and 10 lowest-scoring non-cancer samples from each of the two independent suppliers (Indivumed and MT Group). We excluded samples with a raw number of reads below the $10^{th}$ percentile or above the $90^{th}$ percentile among all samples from the same supplier. We down-sampled each sample without replacement at four ratios: 0.8, 0.6, 0.4, 0.2. We then mixed each downsampled sample with all samples from the same supplier that had a different cancer status, ensuring that the sum of downsample rates equaled 1.0. For example, a 0.6x downsampled cancer sample from Supplier 1 was mixed with all 0.4x downsampled non-cancer samples from Supplier 1. This process resulted in the creation of $10 \times 10 \times 4 \times 2 = 800$ in silico diluted samples. We applied the Orion, XGBoost, ElasticNet, and SVM models without retraining and evaluated the predicted scores (Supplementary Fig. 3).

### Reporting summary
Further information on research design is available in the Nature Portfolio Reporting Summary linked to this article.

## Data availability
The raw small RNA-seq data of the non-small lung cancer and adjacent non-cancer tissues of the cancer genome atlas data were obtained through dbGAP (accession phs000178.v11.p8). Samples used in this study are purchased from commercial sources, were not explicitly consented for data release, and are governed by regulations limiting public release of the raw sequencing data. The count matrices and phenotype information of the validation dataset are available in Zenodo (https://doi.org/10.5281/zenodo.12809652)[58]. Source data are provided with this paper.

## Code availability
Orion model source code is available at https://github.com/exai-oss/orion as well as Zenodo (https://doi.org/10.5281/zenodo.13770567)[59].

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

## Acknowledgements

The results shown here are in part based upon data generated by the TCGA Research Network: https://www.cancer.gov/tcga. This study is funded by Exai Bio Inc. H.G. is an Arc Core Investigator.

## Author contributions

All the listed contributions are based on alphabetical order. B.A., P.A., H.G., F.H., and M.K. designed and planned the study. B.A., T.B.C., H.G., and F.H. defined the oncRNA identities. B.A., H.G., F.H., and M.K. developed the Orion algorithm and M.K. performed the analyses with feedback from B.B., O.E., A.M., P.M., and J.Z. N.C., Y.F., J.K., M.M., J.W., and X.Z. processed and analyzed the datasets. D.C., K.H.C., L.F., A. Hu., R.H., S.K., T.L., D.N, and K.W. collected, curated, and processed the clinical samples. B.A., H.G., F.H., and M.K. wrote the manuscript and incorporated feedback from other authors. B.B., T.B.C., O.E., A. Ha., H.L., A.M., J.Y., and J.Z. provided feedback on the final version of the manuscript.

## Competing interests

The authors are either employees, shareholders, or stock option holders of Exai Bio, Inc. B.A., H.G., F.H., and M.K. have a pending patent application (U.S. Patent "Systems and Methods for Early-Stage Cancer Detection and Subtyping" Application Serial No. 18/636,128 and International Application No. PCT/US24/24682) related to this work.
