## [Transparent Peer Review file · Nature Communications]

Deep generative AI models analyzing circulating orphan non-coding RNAs enable detection of early-stage lung cancer

Corresponding Author: Professor Hani Goodarzi

Version 0:

Reviewer comments:

Reviewer #1

(Remarks to the Author)

Key results

Karimzadeh et al. demonstrate the effectiveness of deep generative AI in accurately detecting lung cancer using orphan non-coding RNA in blood serum. The authors utilize publicly available data from The Cancer Genome Atlas (TCGA) and over 1,050 liquid biopsy samples, well-matched and collected at various units, strengthening the study's reproducibility. The authors employ the Orion model for early cancer detection. The authors aggregated orphan noncoding RNA data into low dimensional representation using VAE, significantly surpassing the performance of other tested methods such as SVM, ElasticNet, and XGBoost. Remarkably, the Orion model achieves a sensitivity of 92% at a specificity of 90%. The paper is even more interesting as oncRNAs are not a standard liquid biopsy material.

Validity

The study is exceptionally well-designed and seamlessly integrates tissue biopsy with liquid biopsy data. The figures presented are compelling and effectively support the authors' claims. The developed model is subject to thorough validation; with an additional comparison of Orion to SVM, ElasticNet, and XGBoost. The authors assess the model's performance concerning triplet margin loss, no reconstruction and no generative sampling. Furthermore, the authors illustrate how AUC and cross-entropy loss vary depending on the number of samples used for training and the number of epochs. The use of SHAP scores to identify specific RNAs that influence the final prediction is another valuable aspect of the study.

Significance

The conclusions of the study are solid, with high potential for clinical implementation. When compared with ctDNA, CTCs, TEPs, miRNAs, or other sources of liquid biopsy material, orphan non-coding RNAs seem a very promising alternative and remain an understudied field. They may spark further research in this direction. The study has many strengths: large sample size, cohorts collected by two independent suppliers, adequate sample-matching, seamless data integration between a publicly available dataset and an in-house dataset, very thorough analysis, multiple validation tests, comparisons with alternative ML approaches, analysis across all cancer stages, analysis with respect to smoking history, liquid histology and model explainability.

Data and methodology

The computational aspects of the study are well-presented. However, the study lacks a detailed description of the wet-lab protocols:

- How was the blood obtained (which tubes were used) and centrifuged?
- What RNA extraction kit was used?
- What sequencing platform was applied, and what were its parameters?

Additionally, the authors do not provide an lncRNA database, raw sequencing data, or code, which contradicts open science principles. In this form, the study is not reproducible.

Analytical approach

The analytical approach in this study is impressive, surpassing that of most manuscripts that combine machine learning with liquid biopsies.

Suggested improvements

Major concerns:

1. Data availability: please provide orphan RNA database, sequencing raw data and code. In the current form, the study is not reproducible.
2. Wet-lab procedures: please provide information on blood tubes used, serum isolation protocol, RNA extraction kit, reverse transcription kit, sequencing platform and its parameters.
3. The authors refer to Karimzadeh et al. 2023a and Karimzadeh et al. 2023a, as well as Karimzadeh et al. 2023c, but all these references are conference abstracts. There is no way of verifying the provided results.
4. Paragraph 2 (page 13) mentions "74 million distinct candidates" – please provide more details.

Minor concerns:

1. Technically, stage III NSCLC, being a locally advanced cancer, does not count as an early stage (paragraph 1, page 2).
2. Please explain the rationale behind setting specificity to 90%. For screening tests, this metric is often expected to be higher.
3. The authors mention 55%–57% sensitivity for the ctDNA assay of Lebow et al. and Cascone et al. but do not state the specificity of these tests. Please provide additional information (paragraph 2, page 2).
4. In the 4th paragraph of page 9, the authors claim Orion could be used as liquid histology beyond cancer. Please provide an example.
5. What was the rationale behind collecting 1,050 samples? Was the study preceded by sample size calculation?
6. How were the 183 resequenced samples used (page 13)?
7. As the authors do not provide any results on minimal residual disease, paragraph 3 (page 9) seems irrelevant. In my opinion it should be removed.
8. Figure 2e – RNA name seems to be missing between VEGFA and OSBPL1A
9. Figure 4 – please also provide % in the confusion matrix.
10. Please rephrase the 3rd sentence on page 8 as it does not sound grammatically correct.

Clarity and context

The study is very clearly explained and easy to follow.

References

It does when it comes to papers of other authors. References co-authored by the first author are not informative.

Your expertise

The review team consisted on two researchers. One is a molecular biologist, head of the Biostatistics core facility and a group leader at a medical university, with over 10 years of experience in the liquid biopsy field. The second researcher is a senior biostatistician, a senior bioinformatician, and a machine learning expert. We believe that together, we do possess the expertise necessary to evaluate this work.

Reviewer #2

(Remarks to the Author)

Deep generative AI models analyzing circulating orphan non-coding RNAs enable accurate detection of early-stage non-small cell lung cancer

Summary

The manuscript presents Orion, a semi-supervised generative model for classifying non-small cell lung cancer (NSCLC) samples from normal samples using orphan non-coding RNAs (oncRNAs) found in blood samples. The manuscript uses smRNA data from 10,403 tumor samples and 679 tumor-adjacent normal samples spanning 32 tissue types to select representative oncRNAs specific to lung cancer. A generalized linear regression model was used to identify ~260,000 lung-tumor-specific oncRNAs differentially abundant in lung tumors compared to normal tissues. Training and evaluation were conducted on a blood serum dataset with 1257 samples from 1050 patients from NSCLC and control cohorts. The Orion model is a variational autoencoder where the encoder learns latent representations for oncRNAs and the endogenous expressed RNAs to estimate library size and complexity. The decoder learns a joint representation of oncRNAs by learning the parameters of a zero-inflated negative binomial distribution.

A triplet margin loss is also used to achieve better separation between normal and cancer samples while accounting for confounders. The supervised task of normal versus cancer samples is evaluated using a cross-entropy loss function. On the validation set, Orion beats XGBoost, SVM, ElasticNet, and KNN models to predict normal versus NSCLC. Finally, the manuscript shows that Orion can distinguish between NSCLC tumor subtypes, squamous cell carcinoma and Adenocarcinoma, with an ROC of 0.75.

Major:

The manuscript reports performance metrics on the training and validation sets, which needs to be clarified since ML performance should always be reported on the test set. The dataset for an ML model is divided into training, validation and test sets, where training data is used to fit the model, validation data is used to estimate hyperparameters and test data is used to evaluate how well the model generalizes to new data. It is also acceptable to report validation set performance to demonstrate that the performance degradation on the test set is not considerable. Training performance indicates how well the model fits the training data. If training continues indefinitely, it will lead to overfitting since the model can start fitting the noise in the training data. It is possible that the manuscript erroneously refers to the validation set as the training set and the test set as the validation set. If this is the case, please fix the terminology. If this is not the case, please report performance on

the test set.

The manuscript aims to develop a robust cancer test using oncRNAs from patient blood samples. For such a test, it is essential to generalize across blood samples from multiple batches, where I use the term batches to refer to sources of technical variation that might arise when collecting oncRNAs from a patient sample. However, the performance is only evaluated on a validation set derived from the same datasets as the training set. This evaluation shows the best-case performance for the Orion model with no real sense of how well the model generalizes to out-of-distribution samples.

The Orion model and associated code are not available for inspection. Without access to the trained models, primary data and any data used to create the figures, it is not possible to verify the claims made in the manuscript. Furthermore, other than listing top SHAP value genes, the manuscript lacks any interpretability analysis for Orion's prediction decisions. A generative model intended for such a significant application, early detection of cancerous samples, requires a much more rigorous probing of the model's decision logic than the currently presented results in the manuscript.

Reviewer #3

(Remarks to the Author)

Version 1:

Reviewer comments:

Reviewer #1

(Remarks to the Author)

The authors have successfully addressed all of our previous concerns, and we are generally satisfied with the revisions made to the manuscript. However, we still face an issue regarding the accessibility of the source data and code. Despite multiple attempts, we are unable to access the GitHub repository, as it requires an authorization code that we do not possess. We kindly request that the authors provide open access to these resources to ensure full transparency and reproducibility of their work.

Reviewer #2

(Remarks to the Author)

The data (<https://doi.org/10.5281/zenodo.12809652>) and code repository (<https://github.com/exai-oss/orion>) links are unavailable for the reviewers. Please provide functional links. Without these data and code, assessing the results presented in the manuscript is not feasible.

Assuming the authors make the data and code available, the review addresses my concerns regarding train/validation/test set separation and interpretation of the Orion model.

Reviewer #3

(Remarks to the Author)

Version 2:

Reviewer comments:

Reviewer #1

(Remarks to the Author)

All the comments and concerns were addressed appropriately, with the exception of the ONCRNAs, which remain difficult to track. While I believe this should be clearly stated, I find the application of generative AI in this paper both inspirational and valuable. Even with the missing information, I believe it is worthy of publication.

Reviewer #2

(Remarks to the Author)

The code and data links are now functional. I have no further concerns.

Reviewer #3

(Remarks to the Author)

REVIEWER COMMENTS

Reviewer #1 (Remarks to the Author): expertise in computational biology and liquid biopsy

Key results

Karimzadeh et al. demonstrate the effectiveness of deep generative AI in accurately detecting lung cancer using orphan non-coding RNA in blood serum. The authors utilize publicly available data from The Cancer Genome Atlas (TCGA) and over 1,050 liquid biopsy samples, well-matched and collected at various units, strengthening the study's reproducibility. The authors employ the Orion model for early cancer detection. The authors aggregated orphan noncoding RNA data into low dimensional representation using VAE, significantly surpassing the performance of other tested methods such as SVM, ElasticNet, and XGBoost. Remarkably, the Orion model achieves a sensitivity of 92% at a specificity of 90%. The paper is even more interesting as oncRNAs are not a standard liquid biopsy material.

Validity

The study is exceptionally well-designed and seamlessly integrates tissue biopsy with liquid biopsy data. The figures presented are compelling and effectively support the authors' claims. The developed model is subject to thorough validation; with an additional comparison of Orion to SVM, ElasticNet, and XGBoost. The authors assess the model's performance concerning triplet margin loss, no reconstruction and no generative sampling. Furthermore, the authors illustrate how AUC and cross-entropy loss vary depending on the number of samples used for training and the number of epochs. The use of SHAP scores to identify specific RNAs that influence the final prediction is another valuable aspect of the study.

Significance

The conclusions of the study are solid, with high potential for clinical implementation. When compared with ctDNA, CTCs, TEPs, miRNAs, or other sources of liquid biopsy material, orphan non-coding RNAs seem a very promising alternative and remain an understudied field. They may spark further research in this direction. The study has many strengths: large sample size, cohorts collected by two independent suppliers, adequate sample-matching, seamless data integration between a publicly available dataset and an in-house dataset, very thorough analysis, multiple validation tests, comparisons with alternative ML approaches, analysis across all cancer stages, analysis with respect to smoking history, liquid histology and model explainability.

Data and methodology

The computational aspects of the study are well-presented. However, the study lacks a detailed description of the wet-lab protocols:

- How was the blood obtained (which tubes were used) and centrifuged?
- What RNA extraction kit was used?
- What sequencing platform was applied, and what were its parameters?

Response to data and methodology concern: We thank the reviewer for the positive comments and noting the lack of detailed wet lab protocols. We have provided additional details under the newly added “Laboratory workflow” section of the manuscript as:

Sample Acquisition and initial processing

Serum samples were acquired from vendors Indivumed (Hamburg, Germany) and MT Group (Los Angeles, CA). All samples were acquired under valid IRB approvals. Serum samples were collected according to standard protocols and frozen at -80C after processing.

RNA extraction and Library Preparation

RNA was extracted from 1 mL of serum using the Quick-cfRNA Serum & Plasma Kit (Zymo). Libraries were prepared using the SMARTer smRNA-Seq Kit for Illumina (Takara) using custom RT primers to incorporate UMIs. Libraries were PCR amplified using Takara SeqAmp DNA Polymerase and indexed during the PCR reaction. Amplified libraries were combined and size selected using either 8% NuPAGE TBE gels (Thermo Fisher) or BluePippin (Sage Science) using 3% Agarose Cassette DF Marker Q3 (Sage Science). Pooled libraries were purified using the Nucleospin Gel and PCR Cleanup Kit (Machery Nagel). Purified libraries were quantified either using the TapeStation using High Sensitivity D1000 Sample Buffer (Agilent) or by using the qPCR-based NEBNext library quant kit for Illumina (NEB) and sequenced on a NextSeq 2000 instrument (Illumina) using P3 flowcells, 50 bp single-end, to an average depth of 19.8 ± 5.8 million mapped reads.

Additionally, the authors do not provide an lncRNA database, raw sequencing data, or code, which contradicts open science principles. In this form, the study is not reproducible.

Response to reproducibility concern: To resolve this limitation, we have provided the Orion code base, Orion model parameters, count data matrices for the validation set, and a notebook reproducing validation set results. We hope that these additions resolve the reviewer’s concerns regarding reproducibility.

Analytical approach

The analytical approach in this study is impressive, surpassing that of most manuscripts that combine machine learning with liquid biopsies.

Suggested improvements

Major concerns:

Comment 1.1. Data availability: please provide orphan RNA database, sequencing raw data and code. In the current form, the study is not reproducible.

Response 1.1. In this revision, we have now provided a repository which includes the Orion source code as well as an example for usage on toy data: <https://github.com/exai-oss/orion>.

We will make this repository and associated data publicly available on Zenodo once the manuscript is accepted for publication (username and password for accessing the repository is provided to the editor for distribution among the reviewers). Additionally, we have provided the model parameters, count matrices for the validation set, and a notebook showing reproduction of model scores for the validation set for the reviewers.

Comment 1.2. Wet-lab procedures: please provide information on blood tubes used, serum isolation protocol, RNA extraction kit, reverse transcription kit, sequencing platform and its parameters.

Response 1.2. We appreciate the reviewer's suggestion regarding a better documentation of wet lab procedures. We have provided additional details under the newly added "Laboratory workflow" section of the manuscript.

Sample Acquisition and initial processing

Serum samples were acquired from vendors Indivumed (Hamburg, Germany) and MT Group (Los Angeles, CA). All samples were acquired under valid IRB approvals. Serum samples were collected according to standard protocols and frozen at -80C after processing.

RNA extraction and Library Preparation

RNA was extracted from 1 mL of serum using the Quick-cfRNA Serum & Plasma Kit (Zymo). Libraries were prepared using the SMARTer smRNA-Seq Kit for Illumina (Takara) using custom RT primers to incorporate UMIs. Libraries were PCR amplified using Takara SeqAmp DNA Polymerase and indexed during the PCR reaction. Amplified libraries were combined and size selected using either 8% NuPAGE TBE gels (Thermo Fisher) or BluePippin (Sage Science) using 3% Agarose Cassette DF Marker Q3 (Sage Science). Pooled libraries were purified using the Nucleospin Gel and PCR Cleanup Kit (Machery Nagel). Purified libraries were quantified either using the TapeStation using High Sensitivity D1000 Sample Buffer (Agilent) or by using the qPCR-based NEBNext library quant kit for Illumina (NEB) and sequenced on a NextSeq 2000 instrument (Illumina) using P3 flowcells, 50 bp single-end, to an average depth of 19.8 ± 5.8 million mapped reads.

Comment 1.3. The authors refer to Karimzadeh et al. 2023a and Karimzadeh et al. 2023a, as well as Karimzadeh et al. 2023c, but all these references are conference abstracts. There is no way of verifying the provided results.

Response 1.3. As suggested by the reviewer, we have removed references to conference abstracts and replaced them with other references when possible.

Comment 1.4. Paragraph 2 (page 13) mentions "74 million distinct candidates" – please provide more details.

Response 1.4. We have updated the sentence (below) to add in additional details on the 74 million distinct candidates.

“This resulted in 74 million distinct (chromosome:start–end:strand), non-overlapping candidate loci having at least one read mapping to a unique position in the genome in at least one sample from 10,403 total number of tumor samples.”

Minor concerns:

Comment 1.1.m. Technically, stage III NSCLC, being a locally advanced cancer, does not count as an early stage (paragraph 1, page 2).

Response 1.1.m. We agree and thank the reviewer for raising this issue. We modified the text to: *“Nationally, only 23% of lung cancer cases are diagnosed **before metastasis (stage I–III)**, when the five-year survival rate is 59%.”*

Comment 1.2.m. Please explain the rationale behind setting specificity to 90%. For screening tests, this metric is often expected to be higher.

Response 1.2.m. Mathios *et al.* 2021 *Nature Com.*

(<https://doi.org/10.1038/s41467-021-24994-w>) chose an 80% specificity for reporting their measurements, which is too permissive for a liquid biopsy application and results in too many instances of over-diagnosis. More recent studies had much lower specificities ranging from 30% to 75% (Hong *et al.* 2024). While multi-cancer detection assays require higher specificities to avoid over-diagnosis, lung cancer screening assays can tolerate lower values of specificity since the target population will undergo additional, more specific secondary screening (Mazzone *et al.* 2024). As a result, we chose a more restrictive cutoff of 90% specificity. Depending on the target population and decision factors outside the scope of this manuscript, a different specificity cutoff may be required for a clinical application. For example, reporting the results for a high specificity such as 99% requires at least 1,000 control samples confirmed with low-dose computed tomography which is outside the scope of this study. To address the reviewer’s comment, in a new supplementary table 2 (revision table 1), we provided the performance reports at higher specificity cutoffs such as 95% and 99% for all methods. These results further show that Orion specificity cutoffs at 95% and 99% are also generalizable to the validation set, while this was not the case for other methods.

Supplementary Table 2: Threshold generalizability and sensitivity of Orion and other methods for the validation set at different cutoffs.

Model	Expected specificity	Specificity	Sensitivity	MCC	F1
Orion	99%	0.98 (0.94–1.00)	0.72 (0.61–0.81)	0.75 (0.66–0.84)	0.82 (0.75–0.89)
Orion	95%	0.94 (0.89–0.98)	0.89 (0.81–0.95)	0.84 (0.76–0.91)	0.90 (0.85–0.95)
Orion	90%	0.87 (0.80–0.93)	0.93 (0.85–0.97)	0.79 (0.70–0.87)	0.88 (0.82–0.93)
ElasticNet	99%	1.00 (0.97–1.00)	0.08 (0.03–0.16)	0.22 (0.13–0.30)	0.15 (0.05–0.25)
ElasticNet	95%	0.99 (0.96–1.00)	0.29 (0.20–0.40)	0.43 (0.33–0.52)	0.45 (0.33–0.56)
ElasticNet	90%	0.98 (0.93–1.00)	0.44 (0.33–0.55)	0.52 (0.41–0.62)	0.59 (0.48–0.69)
XGBoost	99%	1.00 (0.97–1.00)	0.14 (0.08–0.23)	0.30 (0.21–0.39)	0.25 (0.13–0.37)
XGBoost	95%	0.99 (0.96–1.00)	0.53 (0.42–0.64)	0.62 (0.52–0.71)	0.69 (0.59–0.77)
XGBoost	90%	0.98 (0.93–1.00)	0.71 (0.60–0.80)	0.73 (0.64–0.82)	0.81 (0.74–0.87)
KNN	99%	1.00 (0.97–1.00)	0.07 (0.03–0.15)	0.20 (0.12–0.28)	0.13 (0.04–0.22)
KNN	95%	1.00 (0.97–1.00)	0.07 (0.03–0.15)	0.20 (0.09–0.29)	0.13 (0.03–0.23)
KNN	90%	0.98 (0.94–1.00)	0.47 (0.36–0.58)	0.56 (0.47–0.65)	0.63 (0.54–0.72)
SVM-classifier	99%	1.00 (0.97–1.00)	0.07 (0.03–0.15)	0.21 (0.12–0.29)	0.13 (0.05–0.23)
SVM-classifier	95%	0.99 (0.96–1.00)	0.13 (0.07–0.22)	0.26 (0.15–0.35)	0.23 (0.12–0.33)
SVM-classifier	90%	0.99 (0.96–1.00)	0.24 (0.15–0.34)	0.37 (0.26–0.47)	0.37 (0.25–0.49)

Revision Table 1. Threshold generalizability and sensitivity of Orion and other methods for the validation set at different cutoffs.

Comment 1.3.m. The authors mention 55%–57% sensitivity for the ctDNA assay of Lebow et al. and Cascone et al. but do not state the specificity of these tests. Please provide additional information (paragraph 2, page 2).

Response 1.3.m. Thank you for pointing this out. The sensitivity range referenced in Lebow et al. and Cascone et al. correspond to numbers obtained from the tumor-informed ctDNA assay, Signatera (Natera), which has a reported specificity of 99.9% (Signatera Analytical Validation, 2018). We recognize the high specificity achieved using a tumor-informed approach but also emphasize that this type of assay cannot be used for early cancer detection given on its reliance on somatic mutation calls from a primary tumor. We provided these references in the absence of publications using a similar tumor-naive approach.

Since submitting this manuscript, lung cancer detection performance for a ctDNA-based tumor-naive assay has been published. Mazzone et al. (2024) [<https://doi.org/10.1158/2159-8290.CD-24-0519>] report an observed lung cancer detection sensitivity of 84% (95% CI 79%–88%) with a specificity of 53% (95% CI: 45%–61%). The low specificity corresponding to the sensitivity for this assay in this cohort points to the challenges for early lung cancer detection using ctDNA.

We have also added a new sentence describing the newer studies as:

“More recent epigenomic studies report higher sensitivities for lung cancer detection, such a gain usually comes at the cost of the lower specificity. Mazzone et al. (2024), for example, report sensitivity of 84% (95% CI 79%–88%) and specificity of 53% (95% CI: 45%–61%). Similarly, Hong et al. (2024) report sensitivity of 58% (95% CI: 49%–67%) at specificity of 75%

(95% CI: 71%–79%) for stage I and sensitivity of 74% (95% CI: 63%–83%) at specificity of 30% (95% CI: 24%–37%) for stage II lung cancer detection.”

Comment 1.4.m. In the 4th paragraph of page 9, the authors claim Orion could be used as liquid histology beyond cancer. Please provide an example.

Response 1.4.m. The application we are referring to is not beyond cancer per se, but beyond the traditional early detection of cancer to applications such as subtype prediction or minimal-residual detection (MRD). We added a new sentence to elaborate: *“The potential of liquid histology in monitoring tumor subtype transitions, for example, opens new doors for targeted therapy of emerging tumor populations.”*

Comment 1.5.m. What was the rationale behind collecting 1,050 samples? Was the study preceded by sample size calculation?

Response 1.5.m. Previous studies on early detection of lung cancer through liquid biopsy have sample sizes ranging from 288 to 799. For example, Esfahani et al. 2022 have a sample size of 288, Wang et al. 2023 use a total of 601 samples, and Mathios et al. 2021 have a sample size of 799. Our objective was to generate more samples than existing studies to allow us to explore the potential of deep generative models in liquid biopsy applications. We did not explicitly perform sample size calculation for this study, instead we opted for the entirety of the samples that we could acquire from our sources, while ensuring good representation of samples across stages of the disease.

Comment 1.6.m. How were the 183 resequenced samples used (page 13)?

Response 1.6.m. We apologize for the lack of description and also a typo in the reported number (those are 189 samples). The new text reads:

“A subset of the 189 samples with multiple libraries were randomly assigned to the training set. This resulted in 314 samples obtained from 150 patients where each of these patients had at least 2 replicates. During training, all replicates of each sample were allocated either into training or test set during cross-validation to avoid data leakage. Only one of the replicates of each sample was used for reporting the final model performance.”

Comment 1.7.m. As the authors do not provide any results on minimal residual disease, paragraph 3 (page 9) seems irrelevant. In my opinion it should be removed.

Response 1.7.m. We understand the reviewer’s concern. We would like to point out that there are many similarities between early cancer detection and tumor-naive minimal residual disease detection. However, we agree that the manuscript does not discuss any of those similarities, and as the reviewer suggested, we removed this paragraph.

Comment 1.8.m. Figure 2e – RNA name seems to be missing between *VEGFA* and *OSBPL1A*

Response 1.8.m. This is a case where no genes were within 1 Mbp distance of that oncRNA (the last scenario described in the caption).

Comment 1.9.m. Figure 4 – please also provide % in the confusion matrix.

Response 1.9.m. We appreciate the constructive feedback. We added % to the confusion matrix.

Comment 1.10.m. Please rephrase the 3rd sentence on page 8 as it does not sound grammatically correct.

Response 1.10.m. We apologize for the grammatically incorrect sentence. We modified this part of the manuscript as: *“Squamous cell carcinoma transformation of lung adenocarcinoma has been reported to take place spontaneously (Jiang et al., 2019) or after targeted therapy resistance. Such mechanisms of acquired resistance have been reported for epidermal growth factor receptor (EGFR) inhibitors, tyrosine kinase inhibitors (TKIs) (Park et al., 2019), KRAS inhibitors (Tong et al., 2024), and immunotherapies (Hsu et al., 2017).”*

Clarity and context

The study is very clearly explained and easy to follow.

We appreciate the positive feedback regarding the manuscript clarity and context.

References

Comment on references. It does when it comes to papers of other authors. References co-authored by the first author are not informative.

Response to comment on references. We understand the reviewer’s concern regarding the conference abstracts co-authored by the first author. We modified the manuscript and removed references to conference abstracts whenever possible.

Reviewer #1 (Remarks on code availability):

Comment on code availability. We will happily review the code.

Response to comment on code availability. We have provided access to the code repository in the revised manuscript.

Additional comments to reviewer 1.

We appreciate the constructive comments of the reviewer and we believe these comments have helped us better prepare the manuscript's message for the readers of Nature Communications journal. In the attached PDF file, we have clarified all of the changes in the text through strikethrough (text removal) and red text (new additions/corrections). There is also a minor change in the manuscript concerning the validation set performance. This occurred due to incorrect scaling of the validation set count matrices mainly impacting the XGBoost model. This results in minor changes in Figure 2d and supplementary table 1. Other new additions to the manuscript include:

1. Supplementary table 2, showing model performances at different specificity cutoffs
2. Supplementary figure 3, showing qualitative aspects of the model with respect to sequencing depth bias and limit of detection.
3. Supplementary figure 4, showing aspects of model logic including the impact of ablation or permutation of top-SHAP oncRNAs.
4. Code base with example notebooks for training and prediction of Orion
5. Model parameters and count matrices to reproduce Orion model score on the validation set

The source code repository (<https://github.com/exai-oss/orion>) includes the source code of the Orion model and two notebooks showing how to train an Orion model on simulated data or apply an existing Orion model on the validation set datasets.

The dataset repository (<https://github.com/exai-oss/orion-data>) includes count matrices and phenotype information of the validation dataset. This dataset also includes the Orion model parameters to allow reproducibility of the model performance on the validation set.

The reviewer can access these repositories with the following username and password:

Username: exai-review

Password: OrionPaperReview

We noticed that some IP addresses may have issues accessing the repo since GitHub might enforce 2-factor authentication. If this occurs, we request that the reviewer fill this google form and provide a GitHub username so we can provide them access:

<https://forms.gle/NNDRqnUaHvwJJknLA>

We hope these additions address the reviewer's concerns and would allow for publication of our study.

Reviewer #2 (Remarks to the Author): expertise in computational biology

Deep generative AI models analyzing circulating orphan non-coding RNAs enable accurate detection of early-stage non-small cell lung cancer

Summary

The manuscript presents Orion, a semi-supervised generative model for classifying non-small cell lung cancer (NSCLC) samples from normal samples using orphan non-coding RNAs (oncRNAs) found in blood samples. The manuscript uses smRNA data from 10,403 tumor samples and 679 tumor-adjacent normal samples spanning 32 tissue types to select representative oncRNAs specific to lung cancer. A generalized linear regression model was used to identify ~260,000 lung-tumor-specific oncRNAs differentially abundant in lung tumors compared to normal tissues. Training and evaluation were conducted on a blood serum dataset with 1257 samples from 1050 patients from NSCLC and control cohorts. The Orion model is a variational autoencoder where the encoder learns latent representations for oncRNAs and the endogenous expressed RNAs to estimate library size and complexity. The decoder learns a joint representation of oncRNAs by learning the parameters of a zero-inflated negative binomial distribution.

A triplet margin loss is also used to achieve better separation between normal and cancer samples while accounting for confounders. The supervised task of normal versus cancer samples is evaluated using a cross-entropy loss function. On the validation set, Orion beats XGBoost, SVM, ElasticNet, and KNN models to predict normal versus NSCLC. Finally, the manuscript shows that Orion can distinguish between NSCLC tumor subtypes, squamous cell carcinoma and Adenocarcinoma, with an ROC of 0.75.

Major:

Comment 2.1. The manuscript reports performance metrics on the training and validation sets, which needs to be clarified since ML performance should always be reported on the test set. The dataset for an ML model is divided into training, validation and test sets, where training data is used to fit the model, validation data is used to estimate hyperparameters and test data is used to evaluate how well the model generalizes to new data. It is also acceptable to report validation set performance to demonstrate that the performance degradation on the test set is not considerable. Training performance indicates how well the model fits the training data. If training continues indefinitely, it will lead to overfitting since the model can start fitting the noise in the training data. It is possible that the manuscript erroneously refers to the validation set as the training set and the test set as the validation set. If this is the case, please fix the terminology. If this is not the case, please report performance on the test set.

Response to comment 2.1. We thank the reviewer for bringing this point to our attention. There is a mis-match in terminology between the ML field and liquid biopsy field. While in ML, samples are split into train/validation/test, in the liquid biopsy field, they are divided into training (or discovery) and validation set. We are following the liquid biopsy terminology in our paper, and we have now taken steps to make sure our process is clear in the text.

To reiterate, our training process involves a 10-fold CV, where in every fold, 90% of samples are used for training and 10% for tuning (equivalent to “validation split”). Training is done for 300 epochs, and as shown in Fig 1d, the model is not overfitting. It is also possible to implement early stopping here, but in our hands, it was not needed for this dataset. Not using early stopping has allowed us to report performance metrics and ROC curves for the combined cross-validated training.

Once all 10 folds are enumerated, the resulting 50 models (we trained 5 models with different random seeds for each fold of the training dataset) are used to score every sample in the held-out validation set (i.e. “test split”) and the results are averaged into a single score.

This is a scheme previously used in the field, for example, Mathios et al. 2021 train and evaluate the model within their LUCAS cohort through cross-validation (similar to what we do here), and evaluate the model performance on their validation cohort. Esfahani et al. 2022 also use the same training/validation terminology, where the training set performance is a leave-one-out cross-validated measurement and the validation set is the equivalent of ML literature test set. To remain consistent with these other studies of lung cancer liquid biopsy, we prefer to keep our existing nomenclature. If after reading this response, the reviewer still prefers us to modify our nomenclature, we would be happy to do so.

We added the following text to **Results: Description of the Datasets** section to clarify our nomenclature and avoid confusion.

“We used 80% of these samples for model training and evaluation through 10-fold cross-validation. During cross-validation, for each of the 10 folds, we used 90% of the samples (training set) for training 5 models with different random seeds and the remaining 10% of the samples (tuning set) for assessing the cross-validated performance of the model. We used the average score of the 50 models on the held-out 20% of the data (validation set).”

Comment 2.2. The manuscript aims to develop a robust cancer test using oncRNAs from patient blood samples. For such a test, it is essential to **generalize across blood samples from multiple batches**, where I use the term batches to refer to sources of technical variation that might arise when collecting oncRNAs from a patient sample. However, the performance is only evaluated on a validation set derived from the same datasets as the training set. This evaluation shows the best-case performance for the Orion model with no real sense of how well the model generalizes **to out-of-distribution samples**.

Response to comment 2.2. We thank the reviewer for this comment, and we hope our clarification here addresses their concern. As mentioned above, our study closely follows that of others in the field (Mathios et al. 2021, Esfahani et al. 2022). It is important to note the dataset we have here is highly heterogeneous; we have collected samples from two vendors, each contracting tens of blood collection sites. The samples were then processed in four separate studies (or processing batches) spanning data collected for well over a year. A

number of patients were included in multiple studies and we have observed highly concordant model scores (validation set agreement of 0.84 (95% CI 0.67–0.91)). Therefore, our processing variation is intended to be extremely low and our sample sites are diverse to reflect the real-world heterogeneity of patients.

The reviewer's assertion is correct that, in the end, clinical evaluation of our approach requires an independent prospectively collected cohort. However, such a study is beyond the scope of this work. We should also emphasize that, in diagnostics, the goal is not to develop models that generalize to out-of-distribution samples, rather the goal is to ensure samples are within training distribution. This is achieved by developing highly standardized collection and processing protocols. In fact, since machine learning models are known to perform poorly for out-of-distribution (OOD) samples, real-world applications should involve detection of OOD samples to avoid the unexpected consequences of incorrect predictions (<https://doi.org/10.48550/arXiv.2109.14885>).

We have now added the following sentences in the discussion to clarify this: *“As a machine learning method, we do not expect Orion to generalize to out-of-distribution samples. In fact, real-life applications of machine learning models must include detection of out-of-distribution samples to avoid generating spurious predictions [Zadorozhny et al. 2022]. The deviation of Orion loss terms for new samples has the potential of facilitating the identification of out of distribution samples”*.

Also, in response to comment 2.3, we performed some additional experiments (discussed below) that use new control samples sequenced at different sequencing depths or generate new *in silico* samples with a different yet controlled distribution of oncRNAs compared to training data. We hope these additional experiments provide more evidence with respect to model generalizability.

Comment 2.3. The Orion model and associated code are not available for inspection. Without access to the trained models, primary data and any data used to create the figures, it is not possible to verify the claims made in the manuscript. Furthermore, other than listing top SHAP value genes, the manuscript lacks any interpretability analysis for Orion's prediction decisions. A generative model intended for such a significant application, early detection of cancerous samples, **requires a much more rigorous probing of the model's decision logic than the currently presented results in the manuscript.**

Response to comment 2.3. We thank the reviewer for their valuable comment. We performed additional analysis to better visualize Orion's decision logic, bias, and limitations. The new supplementary figures 3–4 (revision figures 1–2).

First, we sought to ensure that the model's decision logic is not biased by the sequencing depth. We sequenced 7 technical replicates from a pool of 3 control samples sourced from

bioIVT (Hicksville, NY) at different sequencing depths. Orion correctly classified these samples as control at different sequencing depths and model scores did not increase as a result of higher sequencing depth (Revision figure 1a–b; supplementary figure 3a–b).

As a detection model, we expected the Orion model's logic to produce lower scores when cancer samples are diluted. To test our hypothesis regarding this aspect, we performed an *in silico* dilution assay, combining sequencing reads of cancer and control samples at different ratios. The results confirmed our hypothesis about the model logic since the scores decreased as a result of *in silico* dilution, while it helped us better understand Orion's limit of detection (Revision figure 1c; supplementary figure 3c).

Furthermore, we also investigated the impact of top high-SHAP oncRNAs by setting them to zero (ablation) or randomizing their values among all samples (permutation). As expected from a detection model, ablation mainly decreases model sensitivity (reducing the score of cancer samples), while permutation decreases model specificity (increasing the score of control samples) as shown in Revision figure 2 (supplementary figure 4).

We have now described these additional experiments and analyses in the revised manuscript.

Revision Figure 1. Orion robustness and limit of detection. (a) Model score of a pool of the sera from individuals without lung cancer sequenced at different depths. Horizontal axis shows 5 different target sequencing depths ranging from 4 to 60 million reads. Vertical axis shows the model score for Orion (ref) and XGBoost (blue) as boxplots. (b) Adjusted R2 of linear models for association of model scores with target sequencing depth for XGBoost (left; blue) and Orion (right; red). Mann-Whitney U-test two-sided p-value was 9.756×10^{-29} and U-statistic was 9,552. (c) Horizontal axis shows the fraction of cancer samples in an *in silico* dilution of cancer and control samples. Vertical axis shows the sensitivity of the assay with respect to the 90% specificity cutoff of each method. Data points correspond to Orion (ref), XGBoost (blue), support vector machine classifier (green), and ElasticNet (purple).

Revision Figure 2. Orion tolerance to ablation and permutation of top-SHAP oncRNAs. Horizontal axis shows the number of oncRNAs with highest SHAP values impacted by the experiment. Vertical axis shows model performance as measured by area under ROC (AUC; red), sensitivity (blue), and specificity (green). Top panel shows the impact of setting the top-SHAP oncRNAs to zero among all validation set samples (ablation experiment). Bottom panel shows the impact of permuting top-SHAP oncRNAs among all samples (permutation experiment).

Reviewer #2 (Remarks on code availability):

Code, trained models, and associated data are not available for inspection.

Response to remarks on code availability. In this revision, we have provided the code, trained models, and count matrices to allow reproduction of the results.

Additional comments to reviewer 2.

We appreciate the constructive comments of the reviewer and we believe these comments have helped us better prepare the manuscript's message for the readers of Nature Communications journal. In the attached PDF file, we have clarified all of the changes in the text through strikethrough (text removal) and red text (new additions/corrections). There is also a minor change in the manuscript concerning the validation set performance. This occurred due to incorrect scaling of the validation set count matrices mainly impacting the XGBoost

model. This results in minor changes in Figure 2d and supplementary table 1. Other new additions to the manuscript include:

1. Supplementary table 2, showing model performances at different specificity cutoffs.
2. Code base with example notebooks for training and prediction of Orion
3. Model parameters and count matrices to reproduce Orion model score on the validation set

We hope these additions address the reviewer's concerns and would allow for publication of our study.

Reviewer #3 (Remarks to the Author): ECR co-review

Additional comments to reviewer 3.

We appreciate the reviewer for participating in the ECR co-review. Their constructive comments have helped us better prepare the manuscript's message for the readers of Nature Communications journal. In the attached PDF file, we have clarified all of the changes in the text through strikethrough (text removal) and red text (new additions/corrections). There is also a minor change in the manuscript concerning the validation set performance. This occurred due to incorrect scaling of the validation set count matrices mainly impacting the XGBoost model. This results in minor changes in Figure 2d and supplementary table 1. Other new additions to the manuscript include:

1. Supplementary table 2, showing model performances at different specificity cutoffs
2. Supplementary figure 3, showing qualitative aspects of the model with respect to sequencing depth bias and limit of detection.
3. Supplementary figure 4, showing aspects of model logic including the impact of ablation or permutation of top-SHAP oncRNAs.
4. Code base with example notebooks for training and prediction of Orion
5. Model parameters and count matrices to reproduce Orion model score on the validation set

The source code repository (<https://github.com/exai-oss/orion>) includes the source code of the Orion model and two notebooks showing how to train an Orion model on simulated data or apply an existing Orion model on the validation set datasets.

The dataset repository (<https://github.com/exai-oss/orion-data>) includes count matrices and phenotype information of the validation dataset. This dataset also includes the Orion model parameters to allow reproducibility of the model performance on the validation set.

The reviewer can access these repositories with the following username and password:

Username: exai-review

Password: OrionPaperReview

We noticed that some IP addresses may have issues accessing the repo since GitHub might enforce 2-factor authentication. If this occurs, we request that the reviewer fill this google form and provide a GitHub username so we can provide them access:

<https://forms.gle/NNDRqnUaHvwJJknLA>

We hope these additions address the reviewer's concerns and would allow for publication of our study.

REVIEWER COMMENTS

Reviewer #1 (Remarks to the Author):

The authors have successfully addressed all of our previous concerns, and we are generally satisfied with the revisions made to the manuscript. However, we still face an issue regarding the accessibility of the source data and code. Despite multiple attempts, we are unable to access the GitHub repository, as it requires an authorization code that we do not possess. We kindly request that the authors provide open access to these resources to ensure full transparency and reproducibility of their work.

Reviewer #1 (Remarks on code availability):

We still cannot assess the code.

Response to reviewer 1.

We thank the reviewer for all of the constructive feedback they have provided. We are delighted to hear that their concerns have been addressed. We apologize for the issues regarding accessibility of the source data and code. We made the repository containing the data (<https://doi.org/10.5281/zenodo.12809652>) as well as the code (<https://github.com/exai-oss/orion>) publicly available.

Reviewer #2 (Remarks to the Author):

The data (<https://doi.org/10.5281/zenodo.12809652>) and code repository (<https://github.com/exai-oss/orion>) links are unavailable for the reviewers. Please provide functional links. Without these data and code, assessing the results presented in the manuscript is not feasible.

Assuming the authors make the data and code available, the review addresses my concerns regarding train/validation/test set separation and interpretation of the Orion model.

Reviewer #2 (Remarks on code availability):

The data (<https://doi.org/10.5281/zenodo.12809652>) and code repository (<https://github.com/exai-oss/orion>) links are unavailable for the reviewers.

Response to reviewer 2.

We thank the reviewer for all of the constructive feedback they have provided. We are glad that we were able to address the reviewer's concerns regarding the dataset and model interpretation. We made the repository containing the data (<https://doi.org/10.5281/zenodo.12809652>) as well as the code (<https://github.com/exai-oss/orion>) publicly available.

Reviewer #3 (Remarks to the Author):

Reviewer #3 (Remarks on code availability):

The quality of the code is good and the code is clear. The instruction provided is clear. I was able to easily rerun the analysis on the provided dataset, yielding results that were consistent with those presented in the manuscript.

Response to reviewer 3.

We thank the reviewer for all of the constructive feedback they have provided. We are delighted to see that the reviewer was able to access the datasets, code, and reproduce our findings. We made the repository containing the data (<https://doi.org/10.5281/zenodo.12809652>) as well as the code (<https://github.com/exai-oss/orion>) publicly available.